# Aligned Novel View Image and Geometry Synthesis via Cross-modal Attention Instillation

Min-Seop Kwak[1][*]  Junho Kim[2]  Sangdoo Yun[2,3]  Dongyoon Han[2]
Taekyung Kim[2]  Seungryong Kim[1][†]  Jin-Hwa Kim[2,3][†]

[1]KAIST AI  [2]NAVER AI Lab  [3]SNU AIIS

## Abstract

We introduce a diffusion-based framework that generates aligned novel view images and geometries via a warping-and-inpainting methodology. Unlike prior methods that require dense posed images or pose-embedded generative models limited to in-domain views, our method leverages off-the-shelf geometry predictors to predict partial geometries viewed from reference images, and formulates novel view synthesis as an inpainting task for both image and geometry. To ensure accurate alignment between the generated image and geometry, we propose cross-**Mo**dal **A**ttention **I**nstillation (**MoAI**) where the attention maps from an image diffusion branch are injected into a parallel geometry diffusion branch during both training and inference. This multi-task approach achieves synergistic effects, facilitating both geometrically robust image synthesis and geometry prediction. We further introduce proximity-based mesh conditioning to reduce erroneous projections and to integrate depth and normal cues to the correspondence conditions. Empirically, our method achieves high-fidelity extrapolative view synthesis, delivers competitive reconstruction under interpolation settings, and produces geometrically aligned point clouds as 3D completion. Project page is available at https://cvlab-kaist.github.io/MoAI/.

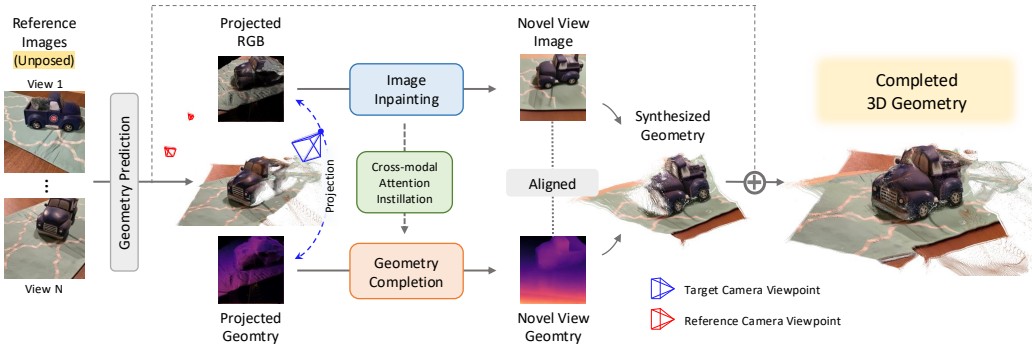

Figure 1: Overview of our diffusion-based framework. From one or more *unposed* reference images, we predict a partial colored point cloud and project it to the target view. Our diffusion model then inpaints missing regions with the cross-**Mo**dal **A**ttention **I**nstillation (**MoAI**), ensuring alignment between image and geometry, resulting in a complete 3D scene.

## 1 Introduction

Novel view synthesis (NVS), the task of reconstructing 3D scenes from sparse 2D reference images, represents a fundamental challenge that requires neural networks to understand and model the underlying 3D structure of a scene from limited 2D observations. Seminal works such as NeRF (Mildenhall et al., 2021) and 3DGS (Kerbl et al., 2023) implicitly model volumetric geometry and appearance by optimizing scene-specific representations to fit individual 3D scenes from reference images. To generalize beyond single-scene optimization, feedforward methods (Wang et al., 2024; Charatan

---

[*]Work done during internship at NAVER AI Lab.
[†]Co-corresponding authors.

et al., 2024; Chen et al., 2024; Ye et al., 2024) have emerged for direct 3D prediction. In addition, the advent of diffusion models (Rombach et al., 2022) has introduced generative NVS methods (Gao et al., 2024; Seo et al., 2024; Ren et al., 2025) that achieve remarkable fidelity in NVS.

However, significant challenges remain. Feedforward methods (Wang et al., 2024; Charatan et al., 2024; Chen et al., 2024; Ye et al., 2024) show high fidelity in interpolative settings by filling the regions visible in reference images, but they lack extrapolation capabilities for synthesizing occluded or unseen areas. Conversely, generative NVS methods (Gao et al., 2024; Seo et al., 2024; Ren et al., 2025), trained with known camera poses, can extrapolate beyond the reference views. However, when conditioned on camera viewpoints underrepresented during training, these models often produce erroneous novel views (Voleti et al., 2024). Consequently, these methods require known reference camera poses, limiting them to the *posed* NVS setting.

Building on warping-and-inpainting methods (Chung et al., 2023; Seo et al., 2024), we propose an alternative framework for multi-view NVS. This framework leverages an off-the-shelf geometry prediction model (Ke et al., 2024; Wang et al., 2024; 2025) to estimate geometry and camera pose from reference images, project this geometry to a target viewpoint, and guide the generative process for plausible NVS. More specifically, we predict geometry from multiple reference views, aggregate and project them to a novel viewpoint, and use this coarse geometric conditioning to guide spatial cross-attention in diffusion networks. By framing NVS as an inpainting problem, our approach synthesizes novel views at arbitrary viewpoints from unposed reference images, while also supporting reconstruction and generation at extrapolative viewpoints. Additionally, we extend this method to novel view geometry synthesis by training a geometry denoising U-Net that inpaints the target geometry from reference views. This offers a key advantage: unlike prior depth-prediction methods Chung et al. (2023); Seo et al. (2024) that suffer from a scale-shift discrepancy between predicted and known reference geometry (Ke et al., 2024), our method ensures geometric alignment by generating as a continuation of the reference geometry.

To encourage synergistic multi-task learning across image and depth modalities, ensuring their alignment, we introduce cross-**Mo**dal **A**ttention **I**nstillation, shortened as **MoAI**, where spatial attention maps from the image denoising network, which implicitly capture cross-view correspondences, are instilled to replace those of the geometry network during both training and inference. More specifically, our design enables to learn from a relatively robust and consistent geometry completion task to regularize image generation through an instilled attention map, while the geometry network leverages rich semantic features from images to improve synthesis quality. Additionally, our proximity-based mesh conditioning incorporates additional geometric cues (depth and normal) into correspondence conditions, interpolating sparse geometry and filtering of erroneous projections.

Our method demonstrates strong extrapolation capabilities for both novel view image and geometry synthesis, resulting in aligned colored point clouds that achieve 3D scene completion (Fig. 1). It achieves state-of-the-art performance in extrapolative settings, while maintaining competitive reconstruction and zero-shot generalization to unseen data.

## 2 RELATED WORK

**Non-generative few-shot NVS.** Neural 3D representations such as NeRF (Mildenhall et al., 2021) and 3DGS (Kerbl et al., 2023) require numerous calibrated views to optimize the neural radiance field effectively. Optimization-based few-shot methods (Jain et al., 2021; Niemeyer et al., 2022; Kim et al., 2022; Kwak et al., 2023) alleviate this issue by tailoring a single 3D scene from sparse views. For instance, DietNeRF (Jain et al., 2021) enforces semantic consistency between rendered images from novel viewpoints and available reference images, while RegNeRF (Niemeyer et al., 2022) regularizes the geometry and appearance of patches from unobserved viewpoints. However, these approaches are unable to generalize beyond individual scenes and are computationally expensive.

Feedforward NVS approaches (Yu et al., 2021; Chen et al., 2024; Wang et al., 2024; Ye et al., 2024; Hong et al., 2023) address few-shot novel view synthesis without per-scene optimization. PixelNeRF (Yu et al., 2021) is among the first to condition a NeRF on image inputs using local CNN features, predicting a novel view image in a feedforward manner. MVSplat (Chen et al., 2024) improves on this by predicting 3D Gaussians from sparse multi-view images with a cost volume for depth estimation, yielding high-quality 3D representations. Subsequent works (Wang et al.,

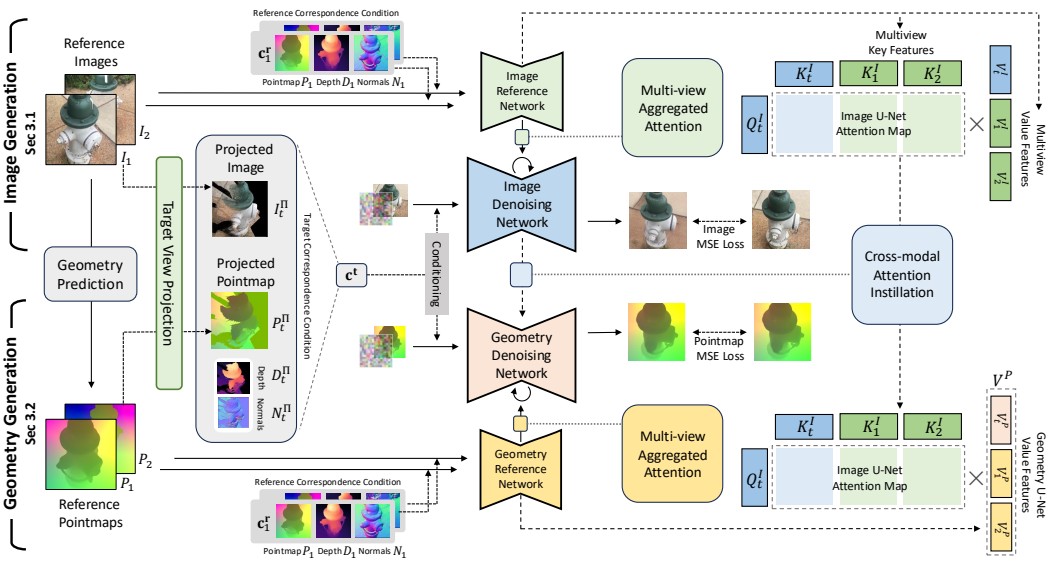

Figure 2: **Training methodology.** Our method conducts cross-modal attention instillation, replacing the spatial attention maps of geometry denoising networks with those of image denoising networks, so that the image generation U-Net learns a more robust representation aligned with the geometry completion task. On the other hand, the geometry prediction networks leverage the rich semantics from image features to enhance geometry completion capability.

2024; Leroy et al., 2024; Hong et al., 2023; Ye et al., 2024) tackle the pose-free scenario, where networks predict novel views from unposed images; for example, DUSt3R (Wang et al., 2024) and MASt3R (Leroy et al., 2024) leverage transformers to output the pointmaps and estimate camera poses, and Noposplat (Ye et al., 2024) jointly predicts 3D representations and poses from sparse inputs. However, these methods generally lack extrapolative capabilities, unable to synthesize novel geometry or appearance in regions unseen or occluded from the reference images.

**Generative few-shot NVS.** Recent diffusion-based NVS approaches (Liu et al., 2023; Voleti et al., 2024; Shi et al., 2023; Gao et al., 2024) leverage their generative capacity to synthesize novel views from one or a few images. Zero123 (Liu et al., 2023) fine-tunes a diffusion model to directly generate a novel viewpoint image for a relative pose from a single image, and ZeroNVS (Sargent et al., 2023) extends upon this for single-view novel view synthesis. Similarly, MVDream (Shi et al., 2023) and CAT3D (Gao et al., 2024) employ spatial cross-attention between the generating viewpoints to achieve consistent novel view synthesis at target viewpoints, while ViewCrafter (Yu et al., 2024) delivers high-fidelity performance by finetuning a video diffusion model. Although these approaches offer strong extrapolative capability, they do not provide explicit geometry for the novel viewpoints, necessitating a separate optimization process on NeRF or 3DGS for full geometry. Moreover, because they receive target view camera pose as a feature embedding, the range of poses they can generate is limited to the training domain, hindering the direct generation of arbitrary novel poses.

**Warping-and-inpainting.** Diffusion models (Rombach et al., 2022) have demonstrated strong inpainting capabilities (Lugmayr et al., 2022), which has motivated their application to novel view synthesis (NVS). LucidDreamer (Chung et al., 2023) leverages off-the-shelf monocular depth estimators (Bhat et al., 2023; Wang et al., 2024; 2025) to extract geometry from a single image, warp it to a target viewpoint, and inpaint missing regions, while GenWarp (Seo et al., 2024) uses predicted geometry as a correspondence signal for implicit inpainting. However, because these methods rely primarily on 2D image inpainting without a comprehensive understanding of 3D structure, they struggle to synthesize scenes with large view differences, particularly in object-centric scenarios where novel view results in warped geometry covering only a small portion of the target viewpoints.

## 3 METHOD

Given $N$ unposed and sparse reference RGB images $\{I_n \in \mathbb{R}^{H \times W \times 3}\}_{n=1}^N$, with height $H$ and weight $W$, the method's objective is the joint prediction of novel view image $I_t$ and pointmap $P_t$ for target viewpoint $\pi_t$, leveraging the diffusion model's generative capabilities for high-fidelity novel view and geometry synthesis. Our method extends the warping-and-inpainting methodology (Leroy et al., 2024; Seo et al., 2024) from single-image to multi-view settings. Importantly, we generalize this strategy from the image domain to geometry, performing geometry completion at the target viewpoint from partial geometry predicted by off-the-shelf models (Sec. 3.1). To ensure alignment between the target image and geometry, we introduce cross-modal attention distillation (Sec. 3.2), a multitask learning that yields synergistic benefits for both modalities. Finally, to handle noise and artifacts in predicted point clouds, we introduce proximity-based mesh conditioning (Sec. 3.3), which prevents erroneous artifacts from degrading generation quality.

### 3.1 NOVEL VIEW IMAGE GENERATION

Our image generation architecture, as shown in the upper section of Fig. 2, consists of two U-Nets: an image reference network and an image denoising network. The image reference network (Hu et al., 2023) extracts semantic reference features from the reference images $\{I_n \in \mathbb{R}^{H \times W \times 3}\}_{n=1}^N$, and the denoising network utilizes the features from the reference network to generate a novel view image.

**Geometry prediction and pointmap projection.** We first leverage an off-the-shelf geometry prediction model (Wang et al., 2024; 2025) to obtain the set of corresponding camera poses $\varphi = \{\pi_n \in \mathbb{R}^{4 \times 4}\}_{n=1}^N$ as well as pointmap $\{P_n \in \mathbb{R}^{H \times W \times 3}\}_{n=1}^N$ from reference images. The pointmap $P_n$ (Wang et al., 2024) is a 2D grid of 3D point coordinates, where each element represents the predicted world coordinate for the given pixel. Next, the pointmaps of the reference images, $P_1, P_2 \ldots P_N$, are interpreted as unordered sets of 3D points then merged into a single point cloud $P$. We then project $P$ onto the target viewpoint $\pi_t$:

$$P_t^{\Pi} = \Pi(P, \pi_t), \quad P = \bigcup_{n=1}^N P_n, \tag{1}$$

resulting in the *projected pointmap* $P_t^{\Pi}$ for the target view $\pi_t$, where $\Pi(\cdot)$ denotes a projection function. When multiple points project onto a single pixel, only the point closest to the target image plane is rendered, as in the standard point cloud rasterization procedure (Seo et al., 2024).

**Pointmap correspondence conditioning.** We leverage the projected pointmap $P_t^{\Pi}$ and the reference view pointmaps $\{P_n\}_{n=1}^N$ as a sparse geometric correspondence condition, which enables the image denoising network to establish the correspondences between the target viewpoint and reference images. Specifically, we first encode $P_t^{\Pi}$ using positional embedding $\mathcal{E}(\cdot)$ and concatenate the resulting Fourier feature $\mathcal{E}(P_t^{\Pi})$ with a binary mask $M_t$, which marks the grid pixels without any projected points. This forms the target correspondence condition $\mathbf{c^t}$ for the image denoising network at target viewpoint $\pi_t$. Similarly, for each reference viewpoint $\pi_n$, we obtain a reference correspondence condition $\mathbf{c_n^r}$ which consists of an embedded reference view pointmap Fourier feature $\mathcal{E}(P_n)$, concatenated with a one-valued tensor mask $\mathbf{1}$, since every grid pixel in the reference view pointmap has a corresponding 3D point due to dense prediction by the off-the-shelf model and therefore marked as 1. As we obtain a reference correspondence condition for every reference image $\{I_n\}_{n=1}^N$:

$$\mathbf{c^t} = [\mathcal{E}(P_t^{\Pi}), M_t], \quad \mathbf{c_n^r} = [\mathcal{E}(P_n), \mathbf{1}], \quad \mathbf{c^r} = \{\mathbf{c_n^r}\}_{n=1}^N. \tag{2}$$

Similar to Hu et al. (2023), these correspondence conditions are first passed through a convolutional network, resulting in target and reference correspondence condition features. The target correspondence condition feature is added to the target image latent feature from the first convolutional layer of the image denoising network, while each reference correspondence condition feature is similarly added to the features of its corresponding reference image within the image reference network. Such conditioning guides the image denoising network in identifying relevant spatial correspondences from multiple reference images to ensure consistency in novel view generation. Notice that instead of providing explicit pixel-to-pixel correlation (*e.g.*, warped pixel coordinates (Seo et al., 2024)) between reference viewpoints and target viewpoint, we directly provide an embedded pointmap as a condition. This design choice allows the model to associate each spatial location in the target image with multiple potential correspondences, providing the reference images for robust reconstruction.

**Aggregated attention.** We conduct an aggregated attention between the target view image features derived from the image denoising network and the reference features produced by the image reference network. We acquire the image key features $K_t^I \in \mathbb{R}^{1 \times C \times (W \times H)}$ and image value features $V_t^I \in \mathbb{R}^{1 \times C \times (W \times H)}$ from the spatial self-attention layers of the image denoising network. These target view image key and value features are concatenated with $N$ image key and value reference features, resulting in the combined image key and value features, $K^I$ and $V^I$, respectively. Then we conduct aggregated attention (Seo et al., 2024) with $K^I$, $V^I$, and the image target view feature $Q_t^I$ as a query:

$$Q^I = Q_t^I, \tag{3}$$

$$K^I = [K_t^I, K_1^I, K_2^I, \dots K_N^I], \tag{4}$$

$$V^I = [V_t^I, V_1^I, V_2^I, \dots V_N^I], \tag{5}$$

where $K^I, V^I \in \mathbb{R}^{(N+1) \times C \times (W \times H)}$. The spatial attention is computed as follows:

$$\text{Attention}(Q^I, K^I, V^I) = \text{Softmax}\left(\frac{Q^I {K^I}^T}{\sqrt{d_k}}\right) V^I, \tag{6}$$

where $d_k$ is the dimensionality of the key features. This design enables the image denoising network to simultaneously perform cross-attention across all reference images and self-attention within the target latents, ensuring a unified novel view synthesis.

## 3.2 ALIGNED NOVEL VIEW GEOMETRY GENERATION

We perform novel view geometry prediction alongside image synthesis using the same architecture as in image generation. The geometry denoising network (U-Net) predicts a target view pointmap, paired with the geometry reference network that receives reference view pointmaps $\{P_n\}_{n=1}^N$. Similar to Ke et al. (2024), our geometry denoising and reference networks are fine-tuned from image denoising U-Nets to predict pointmaps instead of images (*ref.*, Sec. A in the Appendix). As described in the lower half of Fig. 2, the geometry generation part predicts a pointmap for the target viewpoint $\pi_t$, as in the image denoising network, receiving the same conditions, $\mathbf{c^t}$ and $\mathbf{c^r}$. This makes the predicted pointmaps generated by the geometry generation network align with the generated image.

However, we found this naïve approach of giving identical conditions was insufficient for the alignment between the generated image and its geometry. The two modalities exhibit different behaviors when completing void regions, with geometry denoising outperforming image inpainting, as in Fig. 3. This performance difference can be attributed to the more deterministic nature of geometry completion than image generation, as geometric tasks have stronger structural constraints and less inherent ambiguity, resulting in more consistent and robust outcomes.

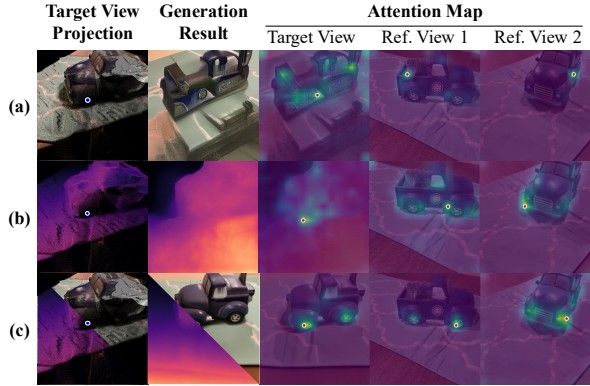

Figure 3: **Effects of cross-modal attention instillation.**

As shown in Fig. 3(a–b), when completing a partially visible wheel, the image denoising (a) fails to establish such correspondences, whereas the geometry prediction (b) correctly attends to other wheel locations for structural reference, producing more realistic completion. Conversely, while the geometry denoising network is better at geometry completion, the spatial attention within these networks struggles to get fine-grained cross-viewpoint correspondences due to the lack of semantic cues, as in (b), where the geometry network's attention is diffusely distributed in comparison to the more focused attention in (a). This complementary relationship motivates a synergistic framework where the geometry network leverages semantic cues from image features via their spatial attention map to achieve more detailed geometry generation, while the structural constraints from geometry completion implicitly guide the image denoising network toward more robust and consistent inpainting.

**Cross-modal attention instillation.** In this light, we propose *cross-modal attention instillation*, where the spatial attention maps for the geometry prediction U-Net are substituted by the attention

maps from the image denoising U-Net to achieve synergistic effects. Specifically, the key and query features extracted from the image denoising U-Net are leveraged by the spatial attention layers of the geometry denoising U-Net. Let $K^I$ and $Q^I$ denote the image key and query features from the image U-Net, respectively, and let $V^P = [V_t^P, V_1^P, V_2^P, \ldots V_N^P]$ represent the geometry value features from the geometry U-Net that generates pointmap $P_t$. The spatial attention is as follows, with $d_k$ as the dimensionality of the key features:

$$\text{Attention}(Q^I, K^I, V^P) = \text{softmax}\left(\frac{Q^I K^{I^T}}{\sqrt{d_k}}\right) V^P, \tag{7}$$

This method offers several key advantages. As shown in Fig. 3, injecting the attention map across the modalities reinforces the alignment between generated images and their geometries. The image denoising U-Net receives deterministic training signals from the geometry completion network, which regularizes its generation process, yielding enhanced consistency and inpainting capability, as demonstrated in Fig. 3 (c). The geometry prediction U-Net also leverages rich semantic features from the image domain to achieve more accurate geometry completion. Additionally, since attention maps serve only as structural cues for aggregating value features, our architecture avoids the detrimental cross-modal feature mixing often observed in the prior work (He et al., 2024).

### 3.3 PROXIMITY-BASED MESH CONDITIONING

The off-the-shelf geometry models (Wang et al., 2024; Ke et al., 2024) typically generate a sparse 3D point cloud with noise and errors, which becomes particularly severe when the target viewpoint deviates significantly from the reference viewpoints. Erroneous projections from the sparse point cloud cause misalignment in the generation process, as the networks cannot differentiate valid from erroneous, and harm the accuracy of generated images and their geometry.

To address these challenges, we propose *proximity-based mesh conditioning*. We convert the sparse point cloud into a mesh representation with the ball-pivoting algorithm (Bernardini et al., 1999), which reduces erroneous projections and yields dense projections for the generation networks. Therefore, instead of employing the naïve projected pointmap $P_t^\Pi$ as the correspondence condition, we utilize the pointmap derived from the projected mesh, denoted $X_t^\Pi$, as our correspondence condition.

Furthermore, we augment the correspondence condition with the mesh's depth and normal map, enabling the network to prioritize reliable correspondences while filtering out noise and erroneous projections. Specifically, we channel-wise concatenate the partial depth map $D_t^\Pi$ and normal map $N_t^\Pi$ acquired from our converted mesh to the correspondence condition embedding. Accordingly, our final correspondence conditions are expressed as follows:

$$\mathbf{c^t} = [\mathcal{E}(X_t^\Pi), D_t^\Pi, N_t^\Pi, M_t], \quad \mathbf{c_n^r} = [\mathcal{E}(X_n), D_n, N_n, \mathbf{1}]. \tag{8}$$

We further refine the conditioning process by applying normal masking to exclude mesh planes whose normals deviate more than 90° from the target viewpoint's direction. These planes typically correspond to surfaces that have been erroneously projected due to the incomplete nature of the acquired geometry, and therefore should be excluded from the correspondence condition. By masking out these areas, we further ensure that the network is not influenced by erroneous or noisy correspondences.

## 4 EXPERIMENT

### 4.1 IMPLEMENTATION AND EXPERIMENTAL DETAILS

For image denoising networks, we initialize from Stable Diffusion 2.1 (Rombach et al., 2022). Reference networks share identical architecture but omit timestep embeddings, serving only for feature extraction. We train on RealEstate10K (Zhou et al., 2018), Co3D (Reizenstein et al., 2021), and MVImgNet (Yu et al., 2023) using pseudo ground-truth geometry from VGGT (Wang et al., 2025). During training, only reference pointmaps are used for warping and proximity-based mesh conditioning. At inference, VGGT predicts reference camera poses and pointmaps for projection to target viewpoints. For geometry denoising networks, we initialize from Marigold (Ke et al., 2024);s normal prediction, finetuning it to generate complete geometry from incomplete warped RGB conditioning rather than target images.

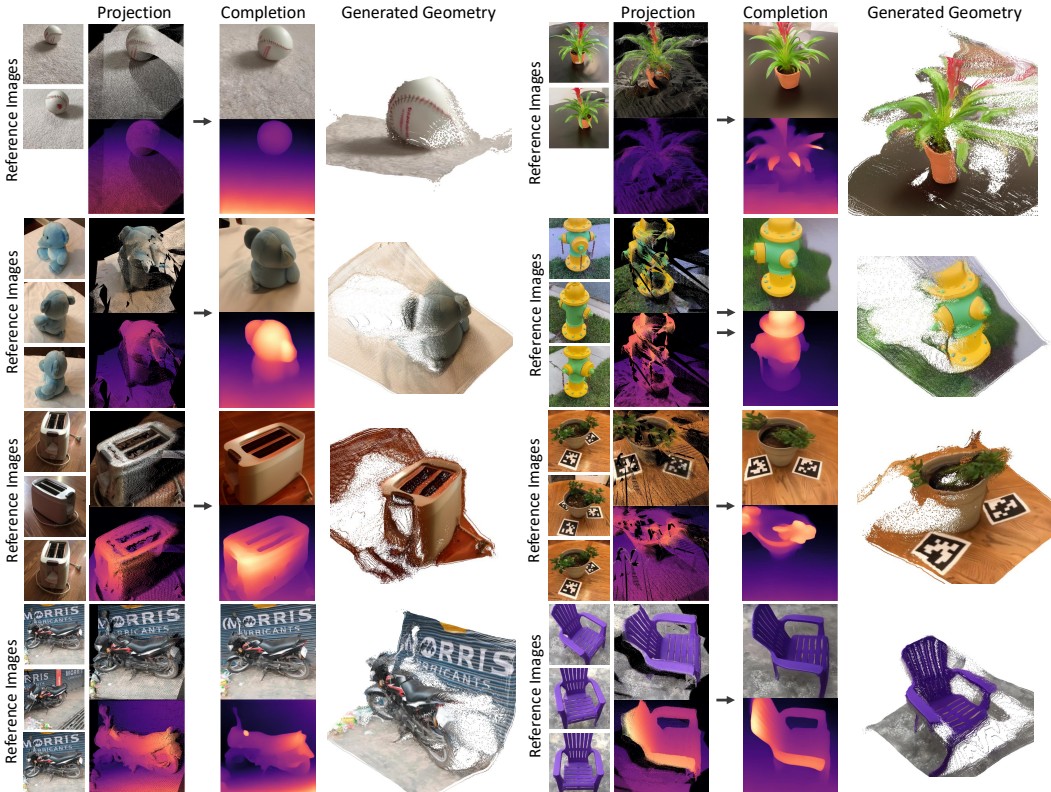

Figure 4: **Qualitative results.** We demonstrate our qualitative results on the Co3D (Reizenstein et al., 2021) dataset, conducting NVS while generating aligned geometry robustly and consistently.

**Extrapolative view setting.** We define extrapolative camera settings as cases when the target camera position $t_t$ lies outside the convex hull of reference camera positions of $\{t_n\}_{n=1}^N$, ignoring viewing direction for simplicity. Alternatively, if $t_t$ cannot be expressed as a convex combination $\Sigma_{n=1}^N \alpha_n t_n$ where $\alpha_n \geq 0$ and $\Sigma_{n=1}^N \alpha_n = 1$. This geometric constraint means that significant portions of the target view may contain regions that are either occluded in reference views or lie beyond the observable scene boundaries, which requires the model to generate plausible scene content based on learned priors about scene structure and appearance to fill in the unknown regions.

## 4.2 EXPERIMENTAL RESULTS

**Results on Co3D.** Figure 4 demonstrates our approach on Co3D, reconstructing novel views with aligned geometry from three reference images. Generated images maintain consistency with references while achieving accurate geometric alignment. Multi-viewpoint pointmap visualizations show well-aligned geometry without scale-and-shift fitting, enabled by formulating depth prediction as completion combined with inpainting. Our denoising network directly generates novel views and geometry at target viewpoints without additional NeRF or 3DGS optimization, overcoming limitations of prior diffusion-based methods (Gao et al., 2024; Wu et al., 2023; Shi et al., 2023).

**Zero-shot results on DTU.** Table 1 and Fig. 5 compare our method against feedforward approaches (PixelSplat (Charatan et al., 2024), MVSplat (Chen et al., 2024), DUSt3R (Wang et al., 2024), NopoSplat (Ye et al., 2024)) using two views, and warping-inpainting methods using single views (LucidDreamer (Chung et al., 2023), GenWarp (Seo et al., 2024)). Evaluation on DTU (Jensen et al., 2014) demonstrates zero-shot generalization capability of our method. For fair comparison, warping methods use identical geometry prediction (VGGT (Wang et al., 2025)). We introduce *extrapolative view* selection sampling the furthest target cameras. Our method achieves state-of-the-art performance in both extrapolative and interpolative settings. Qualitative results show our model effectively filters point cloud artifacts during warping, producing clean images with aligned geometry, while naive inpainting yields artifacts and inconsistent features.

**In-domian results on RealEstate10K.** Table 2 compares our method against feedforward approaches—PixelSplat (Charatan et al., 2024), MVSplat (Chen et al., 2024), DUSt3R (Wang et al.,

Table 1: **Zero-shot quantitative comparison**. We compare our model to existing feedforward NVS methods (2-view setting) and warping-and-inpainting methods (1-view setting) on DTU Zhou et al. (2018) dataset, which is zero-shot setting for all the models. Our method shows superior performance in both extrapolative and original (interpolative) setting in both single-view and stereo-view settings.

| Views | Method | Pose-free | Extrapolative View | | | Interpolative View | | |
|---|---|---|---|---|---|---|---|---|
| | | | PSNR↑ | SSIM↑ | LPIPS↓ | PSNR↑ | SSIM↑ | LPIPS↓ |
| 2-view | PixelSplat (Charatan et al., 2024) | ✗ | 14.66 | 0.517 | 0.334 | 12.75 | 0.329 | 0.637 |
| | MVSplat (Chen et al., 2024) | ✗ | 12.22 | 0.416 | 0.423 | 13.94 | 0.473 | 0.385 |
| | NoPoSplat (Ye et al., 2024) | ✓ | 13.58 | 0.393 | 0.545 | 14.04 | 0.414 | 0.530 |
| | **Ours (2-view)** | ✓ | **15.58** | **0.615** | **0.184** | **16.58** | **0.643** | **0.152** |
| 1-view | LucidDreamer (Chung et al., 2023) | ✓ | 11.14 | 0.423 | 0.440 | 12.09 | 0.481 | 0.419 |
| | GenWarp (Seo et al., 2024) | ✓ | 9.85 | 0.315 | 0.527 | 9.54 | 0.298 | 0.538 |
| | **Ours (1-view)** | ✓ | **15.56** | **0.609** | **0.184** | **14.58** | **0.529** | **0.202** |

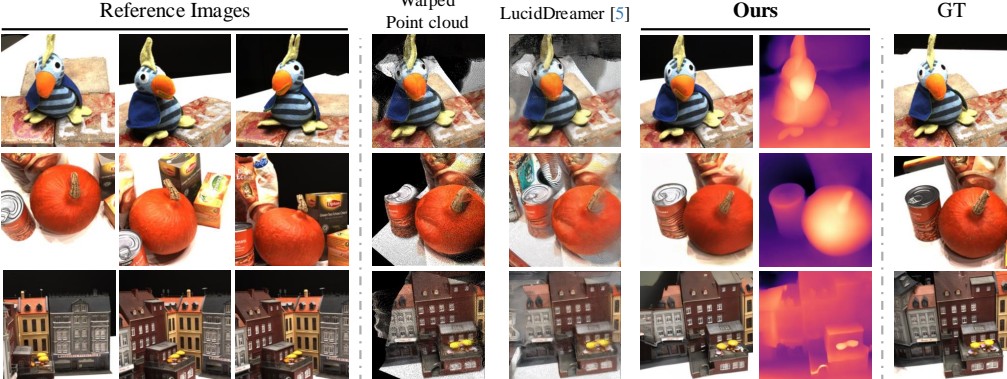

Figure 5: **Qualitative comparison with inpainting method on DTU (Zhou et al., 2018) dataset.** Our qualitative comparison with the naive warping-and-inpainting method demonstrates our model's zero-shot generalization capabilities to unseen data, as well as its ability to robustly handle erroneous warped geometries for geometrically consistent generation.

Table 2: **In-domain comparison**. We compare our model to existing feedforward NVS methods in Realestate10K (Zhou et al., 2018) dataset, our method being superior in extrapolative setting.

| Method | Pose-free | Extrapolative View | | | Interpolative View | | |
|---|---|---|---|---|---|---|---|
| | | PSNR↑ | SSIM↑ | LPIPS↓ | PSNR↑ | SSIM↑ | LPIPS↓ |
| PixelSplat (Charatan et al., 2024) | ✗ | 14.01 | 0.582 | 0.384 | 23.85 | 0.806 | 0.185 |
| MVSplat (Chen et al., 2024) | ✗ | 12.13 | 0.534 | 0.380 | 23.98 | 0.811 | 0.176 |
| NoPoSplat (Ye et al., 2024) | ✓ | 14.36 | 0.538 | 0.389 | **25.03** | **0.838** | 0.160 |
| **Ours** | ✓ | **17.41** | **0.614** | **0.229** | 24.23 | 0.820 | **0.088** |

2024), and NopoSplat (Ye et al., 2024)—on RealEstate10K (Zhou et al., 2018). We evaluate interpolative and extrapolative conditions using two views. For extrapolation, we sample references from the latter third and targets from the first third of video frames. Our model outperforms feedforward methods in extrapolative settings while maintaining competitive interpolative performance. Figure 6 shows conventional models struggle with large missing areas due to limited generative capabilities, whereas our approach effectively infers missing geometry and generates plausible imagery while preserving fidelity. We realistically inpaint partially visible objects with well-aligned geometry, attributed to geometric awareness from attention distillation regularizing structural attention maps.

In Fig 7, we additionally compare our method against recent large model-based novel view synthesis methods, namely LVSM (Jin et al., 2024), ZeroNVS (Sargent et al., 2023), and ViewCrafter (Yu et al., 2024). As shown in the figure, our method excels in extrapolative scenarios with significant unobserved regions. While LVSM accurately reconstructs areas with geometric overlap, it fails beyond observable boundaries, producing blurry content in occluded areas. ZeroNVS produces plausible novel views but lacks fidelity and requires manual specification of field-of-view, elevation, and content scale for each scene, with incorrect values leading to convergence failure. More critically, it requires approximately 2+ hours per scene for NeRF distillation via Score Distillation Sampling. ViewCrafter, evaluated with both 16-frame and 25-frame models, produces geometrically degraded

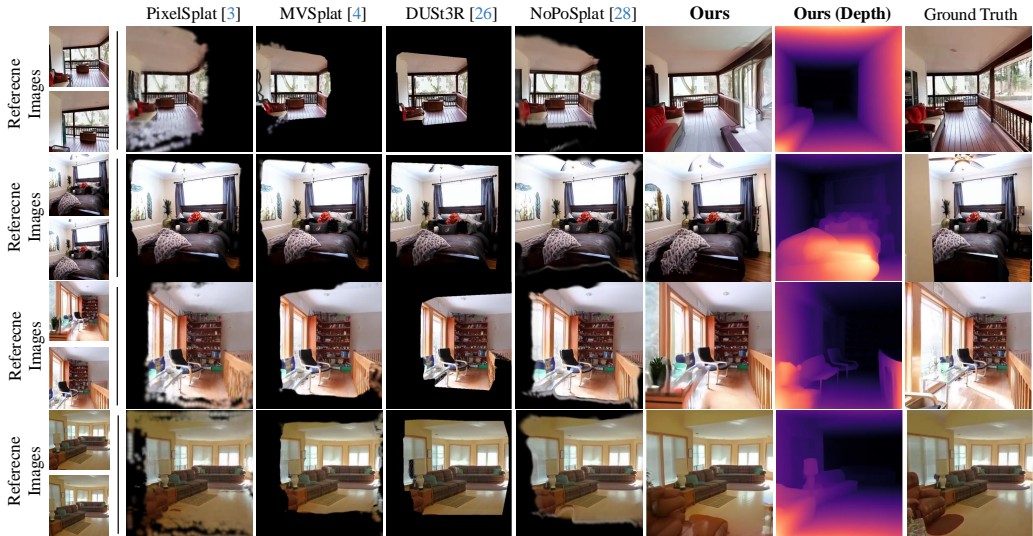

Figure 6: **Qualitative comparison on extrapolative setting.** Our qualitative comparison of previous approaches demonstrates our model's extrapolative capabilities to plausibly generate locations not seen in reference images while reconstructing faithfully the known regions.

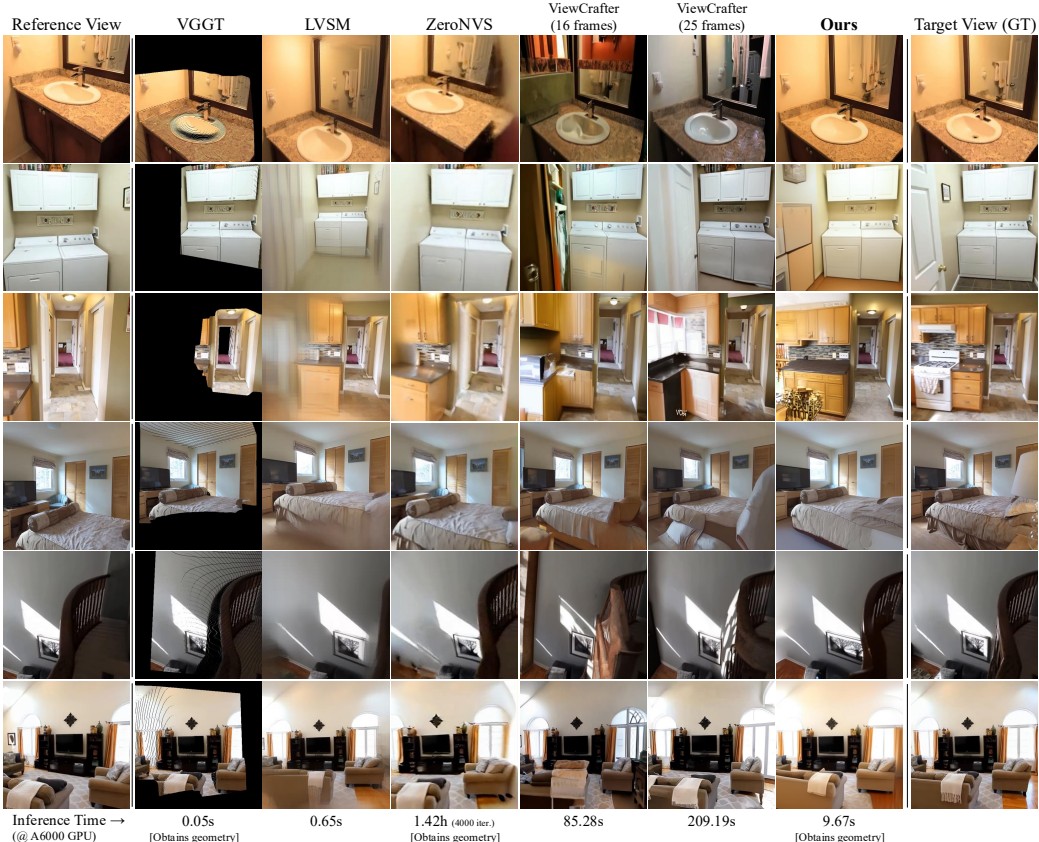

Figure 7: **Qualitative comparison with large model-based NVS methods.** Qualitative comparison of our method to previous approaches demonstrates our method's superior capabilities in conducting geometrically coherent image novel view synthesis with relatively short inference time.

imagery and artifacts under extrapolative settings, requiring 209.19 seconds for 25 frames on an A6000 GPU. In contrast, our generative approach leverages learned priors to synthesize plausible content for missing regions in only 9.67 seconds on average, maintaining high-fidelity detail and

geometric consistency. This demonstrates our model's competitive performance in both quality and efficiency in comparison to current large model-based novel view synthesis methods.

### 4.3 ABLATION

We conduct quantitative ablation studies on our method using the RealEstate10K dataset under the extrapolative setting. Our results in Table 3 demonstrate the contribution of each component. The baseline (a) receives no geometric conditioning, while (b) introduces naive pointmap conditioning. When we add proximity-

Table 3: **Ablation study**. We demonstrate how each of our components contributes to enhanced performance in novel view synthesis.

| Components | PSNR↑ | SSIM↑ | LPIPS↓ |
|---|---|---|---|
| (a) Baseline | 16.55 | 0.559 | 0.260 |
| (b) + Pointmap condition | 16.93 | 0.594 | 0.243 |
| (c) + Proximity-based mesh | 17.01 | 0.601 | 0.238 |
| (d) **+ Cross-modal instillation** | **17.41** | **0.614** | **0.229** |

based mesh conditioning (c) and cross-modal attention distillation (d), we observe progressive performance improvements. We provide additional qualitative results in Fig. 15 of our Appendix.

**Analysis on the number of input viewpoints.** As our model conducts aggregated attention to generate novel views from reference images, it can receive an arbitrary number of input viewpoints for generation. To demonstrate this, in Table 4 and Fig. 18 of our Appendix, we increase the number of reference viewpoints for a model trained at 2-viewpoint setting and analyze its effects in both image quality and geometric accuracy: the results demonstrate even without being trained on the given number of inputs, our model benefits strongly from additional viewpoints, showing the generalization capability of our aggregated attention architecture to various number of input reference viewpoints.

Table 4: **Quantitative analysis regarding number of input viewpoints**. We demonstrate improved performance with additional viewpoints at inference for both image and geometry generation, despite training only on two-view settings.

| Method | Image | | | Geometry | | Geometry (Recon) | | Geometry (Inpainting) | |
|---|---|---|---|---|---|---|---|---|---|
| | PSNR↑ | SSIM↑ | LPIPS↓ | Abs.Rel↓ | $\delta_{1.25}$ ↑ | Abs.Rel↓ | $\delta_{1.25}$ ↑ | Abs.Rel↓ | $\delta_{1.25}$ ↑ |
| 2-view | 17.41 | 0.615 | 0.230 | 0.196 | 0.715 | 0.152 | 0.819 | 0.308 | 0.531 |
| 3-view | 20.02 | 0.700 | 0.146 | 0.143 | 0.788 | 0.151 | **0.849** | 0.304 | **0.598** |
| 4-view | **20.08** | **0.701** | **0.144** | **0.140** | 0.787 | **0.113** | 0.846 | 0.311 | 0.594 |

Table 5: **Robustness to Gaussian perturbation**. Our model shows robustness against Gaussian perturbation to the correspondence condition.

| Perturbation Level | PSNR↑ | SSIM↑ | LPIPS↓ |
|---|---|---|---|
| No noise | 15.580 | 0.615 | 0.184 |
| Gaussian perturb. ($\sigma = 3\%$) | 14.778 | 0.520 | 0.213 |
| Gaussian perturb. ($\sigma = 6\%$) | 14.501 | 0.507 | 0.225 |
| Gaussian perturb. ($\sigma = 10\%$) | 14.129 | 0.487 | 0.239 |
| Gaussian perturb. ($\sigma = 15\%$) | 13.726 | 0.465 | 0.262 |

Table 6: **Robustness to increase sparsity**. Our model shows robustness against increased sparsity in the correspondence condition.

| Masking Level | PSNR↑ | SSIM↑ | LPIPS↓ |
|---|---|---|---|
| No masking | 15.580 | 0.615 | 0.184 |
| 30% masking | 14.683 | 0.577 | 0.223 |
| 50% masking | 13.610 | 0.468 | 0.272 |
| 80% masking | 13.000 | 0.436 | 0.317 |

### 4.4 ANALYSIS ON ROBUSTNESS AGAINST EXTERNAL GEOMETRY PREDICTORS

We demonstrate our model's robustness to errors and artifacts from external geometry predictors by adding various perturbations to the correspondence conditions. First, we apply varying levels of Gaussian noise to predicted pointmap locations and show that our model exhibits minimal performance degradation even under high noise levels (Table 5). Second, we randomly mask points from the predicted pointmap to simulate sparse geometry, finding that our model maintains strong performance despite high sparsity in the correspondence condition (Table 6). Together, these results confirm that our diffusion-based framework's generative capability effectively prevents error propagation from external priors, maintaining our pose-free methodology without compromising output quality. We provide qualitative evaluations of the same experiments in Fig. 10 and Fig. 11 of our Appendix.

### 5 CONCLUSION

We propose a novel few-shot novel-view synthesis method that overcomes the limitations of existing approaches by jointly predicting images and geometry. By integrating cross-modal attention sharing and geometry-aware correspondence conditioning into a warping-and-inpainting framework, our method leverages deterministic cues from geometry completion to regularize the generation process, producing consistent, high-quality novel views even in challenging extrapolative scenarios.

ACKNOWLEDGEMENT

This research was supported by Institute of Information & communications Technology Planning & Evaluation (IITP) grant funded by the Korea government (MSIT) (RS-2019-II190075, RS-2024-00509279, RS-2025-II212068, RS-2023-00227592, RS-2025-02214479, RS-2024-00457882, RS-2025-25441838, RS-2025-25441838, RS-2025-02214479, RS-2025-02217259) and the Culture, Sports, and Tourism R&D Program through the Korea Creative Content Agency grant funded by the Ministryof Culture, Sports and Tourism (RS-2024-00345025, RS-2024-00333068, RS-2023-00222280, RS-2023-00266509), and National Research Foundation of Korea (RS-2024-00346597).

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

# APPENDIX

## A ADDITIONAL DETAILS

### A.1 TRAINING DETAILS

In our training procedure, we initialize the image denoising U-Net from Stable Diffusion 2.1 and fine-tune on RealEstate10K Zhou et al. (2018), Co3D Reizenstein et al. (2021), and MVImgNet Yu et al. (2023). Reference networks share identical architecture and initial weights with the denoising U-Net but exclude timestep embeddings, serving solely for semantic feature extraction. Our model is trained using a batch size of 1 with gradient accumulation, employing mixed-precision training (bf16) to optimize memory usage and computational efficiency. We utilize 8-bit Adam optimizer with standard hyperparameters: $\beta_1 = 0.9$, $\beta_2 = 0.999$, weight decay of $1\times10^{-2}$, and $\epsilon = 1\times10^{-8}$. The learning rate is set to $1\times10$ with a constant scheduler and minimal warmup of 1 step.

For multi-view training, we sample 4 viewpoints total with 3 reference views and 1 target view (index 1), maintaining a temporal margin of 30 frames between reference and target images to ensure sufficient viewpoint diversity. All training images are resized to 512×512 resolution. We employ the xFormers memory-efficient attention mechanisms to handle the computational demands of multi-view aggregated attention. The noise scheduler follows a scaled linear beta schedule with 1000 training timesteps, $\beta_{start} = 0.00085$, $\beta_{end} = 0.012$, and a step offset of 1, without sample clipping. Validation is performed every 2000 training steps to monitor convergence and prevent overfitting.

### A.2 COMPUTATIONAL EFFICIENCY

Despite employing a dual-branch architecture with multi-view attention mechanisms, our model maintains competitive inference speed and memory efficiency. For joint 2-view image and geometry generation, our method requires 9.81 seconds on an A6000 GPU with 28GB memory consumption. Notably, cross-modal attention instillation contributes to reduced memory overhead: the geometry branch reuses pre-computed attention maps from the image branch rather than computing separate attention operations, thereby saving both memory and computation. Furthermore, our image branch can operate independently for image-only novel view synthesis, requiring only 14GB memory and 4.3 seconds inference time—comparable to standard diffusion models. This flexibility allows users to trade off between full geometric consistency (dual-branch mode) and faster image-only generation depending on application requirements.

### A.3 IMPLEMENTATION DETAILS

Geometry for correspondence conditioning is generated using VGGT Wang et al. (2025), with only reference view pointmaps used for warping and proximity-based mesh conditioning. Ground truth geometry for MSE loss supervision is also predicted by VGGT using target images, enabling the model to maintain scale-and-shift alignment with reference geometry automatically. This approach allows exact reconstruction of known geometry while completing unknown regions in alignment with reference observations.

To accelerate and stabilize training, we separately fine-tune the pointmap denoising U-Net and its reference network, initializing from Marigold Ke et al. (2024)'s normal prediction weights with identical multiview aggregated attention as the image U-Net. This separate initialization enables both branches to learn robust independent representations before cross-modal attention distillation, where geometry networks benefit from deterministic image cues, significantly improving prediction consistency. However, as discussed in our main paper, geometry-only training cannot achieve perfectly aligned predictions, necessitating cross-modal attention distillation for optimal performance.

**Camera-space pointmap normalization.** A key insight in our approach involves normalizing coordinate pointmaps through a camera-centric transformation that substantially improves model performance. Specifically, we apply the target viewpoint's world-to-camera matrix to convert all predicted pointmap coordinate values into the target camera's local coordinate system. This pre-processing strategy addresses a critical issue in multi-view geometric learning: when coordinate pointmaps retain their original world-space values, the model must simultaneously learn to handle

dramatic variations in absolute coordinates while capturing subtle geometric relationships. Such dual complexity often hampers training convergence and degrades synthesis quality.

Our camera-space transformation eliminates this burden by establishing a unified coordinate frame centered on the target view. Within this normalized space, all geometric information is expressed relative to the target camera's position and orientation, creating a more conducive learning environment. The model can then dedicate its representational capacity entirely to understanding the geometric correspondence patterns between reference and target configurations, without being distracted by irrelevant absolute positioning. This coordinate system alignment also ensures numerical consistency across training samples, preventing gradient instabilities that can arise from extreme coordinate ranges. The resulting geometric conditioning leads to more stable training dynamics and enhanced ability to synthesize geometrically plausible novel views across diverse camera poses and scene configurations.

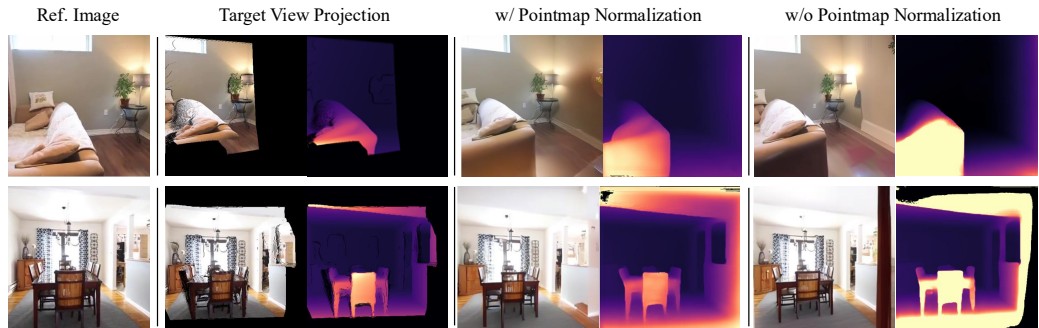

Figure 8: **Ablation on pointmap normalization.** Ablation study comparing synthesis results with and without camera-space pointmap normalization. Normalization significantly improves geometric consistency, boundary sharpness, and geometric alignment with projected geometry.

To validate the effectiveness of our pointmap normalization strategy, we conduct a qualitative ablation study comparing synthesis results with and without camera-space coordinate transformation. Our experimental results in Fig. 8 demonstrates that without normalization, the model struggles to maintain geometric consistency across different viewpoints, producing artifacts such as distorted object boundaries, inconsistent depth relationships, and misaligned features between generated images and their corresponding geometry.

## B  ADDITIONAL RESULTS

### B.1  ADDITIONAL COMPARISON AGAINST LARGE MODELS

We additionally compare against VGGT (Wang et al., 2025), LVSM (Jin et al., 2024) and ViewCrafter (Yu et al., 2024), large model-based novel view synthesis models. As shown in Fig. 9, our method excels in extrapolative scenarios with significant unobserved regions. LVSM fails beyond observable boundaries, producing blurry occluded content, while ViewCrafter generates geometrically degraded imagery under extrapolative settings and requires 209.19 seconds for 25 frames on an A6000 GPU. Our approach synthesizes plausible content in only 9.67 seconds on average, maintaining high-fidelity detail and geometric consistency, demonstrating superior quality and efficiency.

**Quantitative evaluation.** To further validate our model's generalization capability, we conduct a quantitative evaluation on the DTU dataset under extrapolative view settings in Table 7. We compare against ViewCrafter and LVSM under both single-view and two-view settings. Results demonstrate that our model achieves superior performance across all metrics in both configu-

Table 7: **Comparison to large model baselines**. We quantitatively compare our model against recent large-scale models.

| Methods | View setting | PSNR↑ | SSIM↑ | LPIPS↓ |
|---|---|---|---|---|
| ViewCrafter | 1-view | 14.04 | 0.390 | 0.332 |
| Ours (Single-view) | 1-view | 15.56 | 0.609 | 0.184 |
| LVSM | 2-view | 15.23 | 0.499 | 0.415 |
| **Ours (Stereo-view)** | 2-view | **15.58** | **0.615** | **0.184** |

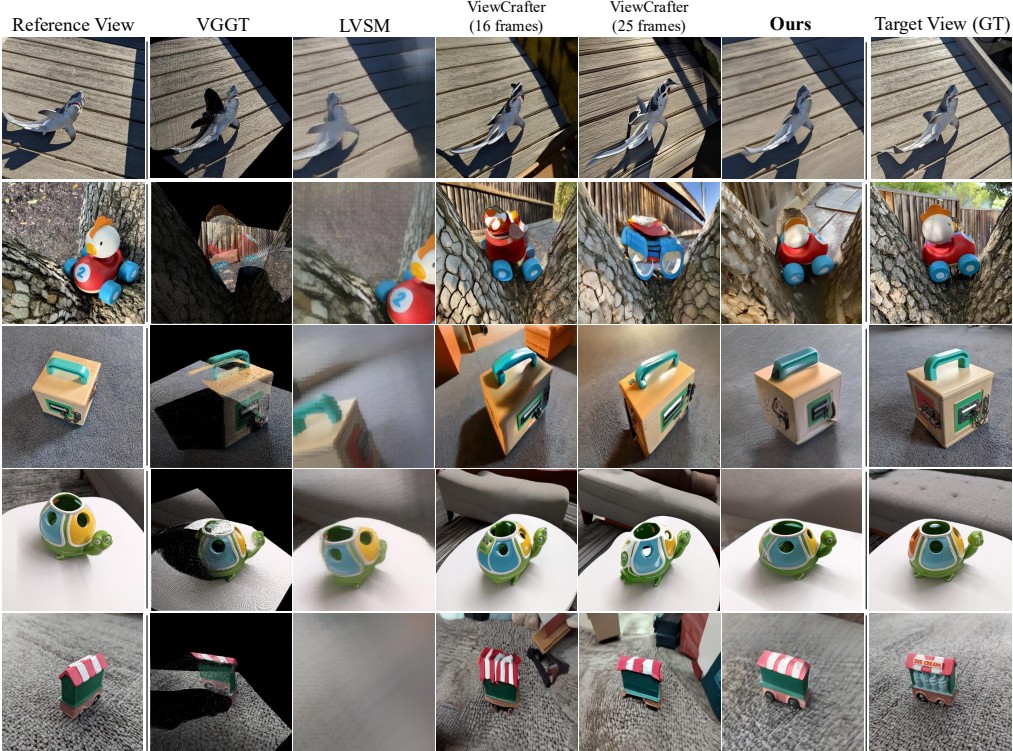

Figure 9: **Qualitative comparison with large models on Navi (Jampani et al., 2023) Dataset.** Our model shows competitive performance against large model-based novel view synthesis models.

rations: under single-view input, our model outperforms ViewCrafter by +1.52 dB PSNR, +0.219 SSIM, and 0.148 LPIPS; under two-view input, our model surpasses LVSM by +0.32 dB PSNR, +0.116 SSIM, and 0.231 LPIPS. These quantitative results on DTU corroborate our qualitative findings on RealEstate10k and Navi datasets, confirming our model's competitive performance against state-of-the-art methods across diverse benchmarks.

### B.2 ROBUSTNESS TO PERTURBATIONS IN CORRESPONDENCE CONDITION

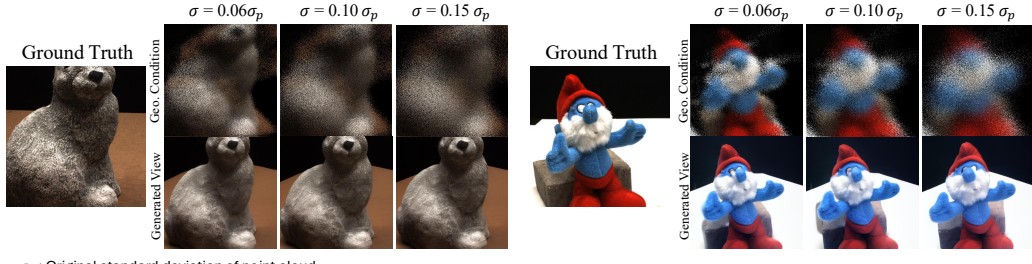

$\sigma_p$ : Original standard deviation of point cloud

Figure 10: **Robustness experiments regarding geometric noise.** Visualization of MoAI's robustness against noise added to the predicted geometry.

In Fig. 10, we present qualitative experiments examining robustness to geometric noise in the predicted geometry. We apply varying levels of Gaussian noise (standard deviations of 6%, 10%, and 15% relative to the standard deviation of reference point cloud coordinates) to the point locations in the predicted pointmaps and use these noisy pointmaps as correspondence conditions. Despite high noise levels, our model exhibits minimal performance degradation, demonstrating strong robustness against errors and artifacts in the predicted geometry. In Fig. 11, we provide similar experiments regarding the sparsity of the predicted geometry: we randomly mask out a certain percentage of points (30%, 50% and 80%) from the predicted pointmap, and use this sparse pointmap as the

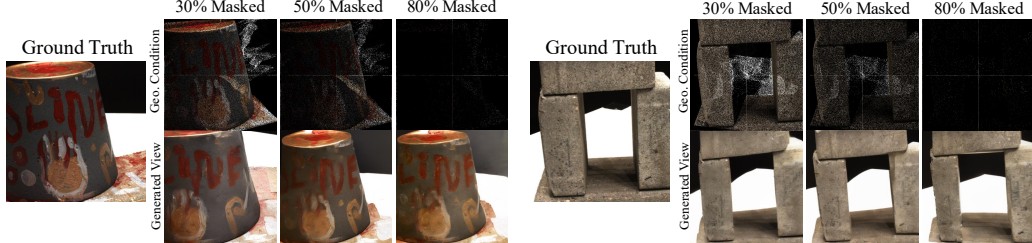

Figure 11: **Robustness experiments regarding sparsity.** Visualization of MoAI's robustness against increased sparsity of predicted geometry.

correspondence condition. Despite high sparsity in the correspondence condition, our model shows minimal performance degradation even at 80% masking scenario, demonstrating robustness against sparsity in predicted geometry. These results confirm that our framework's generative approach effectively mitigates errors from external priors, maintaining our pose-free methodology without compromising output quality.

### B.3 Qualitative evaluation with in-the-wild data

We provide additional results demonstrating our model's strong generalization to unseen domains, including in-the-wild and urban data. We conducted experiments on two new datasets from different domains: the CityScapes (Cordts et al., 2016) dataset, containing urban street-view multi-view images; the MegaDepth (Li & Snavely, 2018) dataset. None of these datasets was used during training, thereby validating our model's generalization capabilities. Fig. 12 presents qualitative results demonstrating high-quality novel view synthesis and geometry reconstruction on in-the-wild data, confirming strong generative performance across diverse domains.

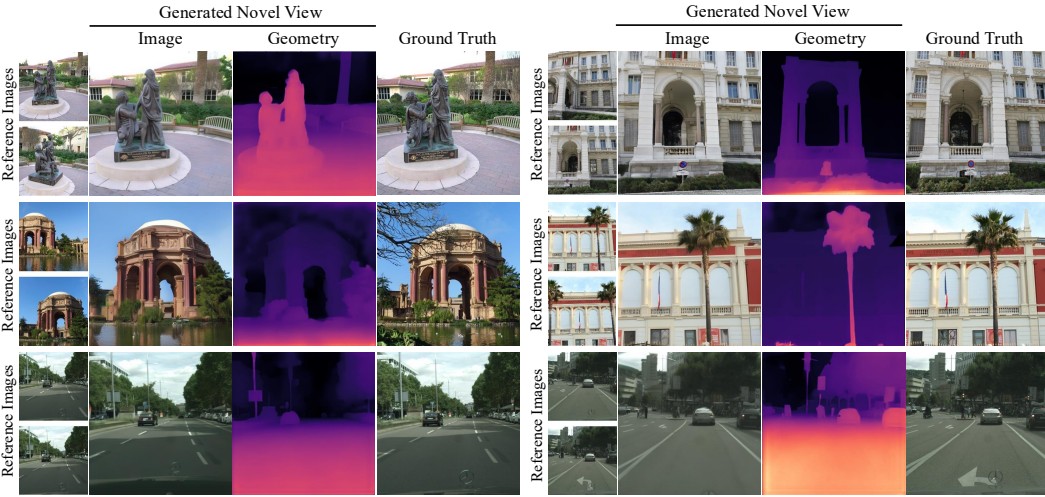

Figure 12: **Qualitative results on in-the-wild / urban data.** We provide generalization results on unseen in-the-wild / urban datasets (MegaDepth, CityScapes), for which our model is capable of conducting high-fidelity novel view synthesis.

### B.4 Evaluation regarding geometric consistency in occlusion and shading

Fig. 13, in addition to Fig. 4, demonstrates extrapolative view cases where our model generates target views more than 90° away from the reference viewpoints, requiring generation of completely unseen regions. Our model maintains physical consistency when extrapolating beyond 90° view differences, including correct shading and occlusion handling.

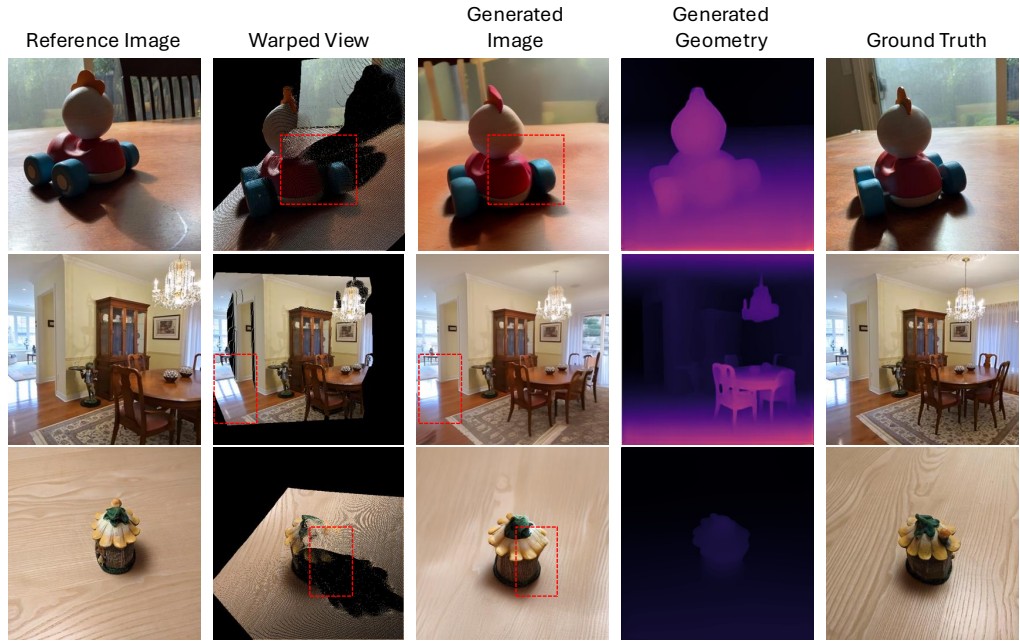

Figure 13: **Occlusion handling in extrapolative viewpoints.** We demonstrate the performance of our model under occlusion / lighting / shadow handling in extrapolative (greater than 90 degrees) target camera pose.

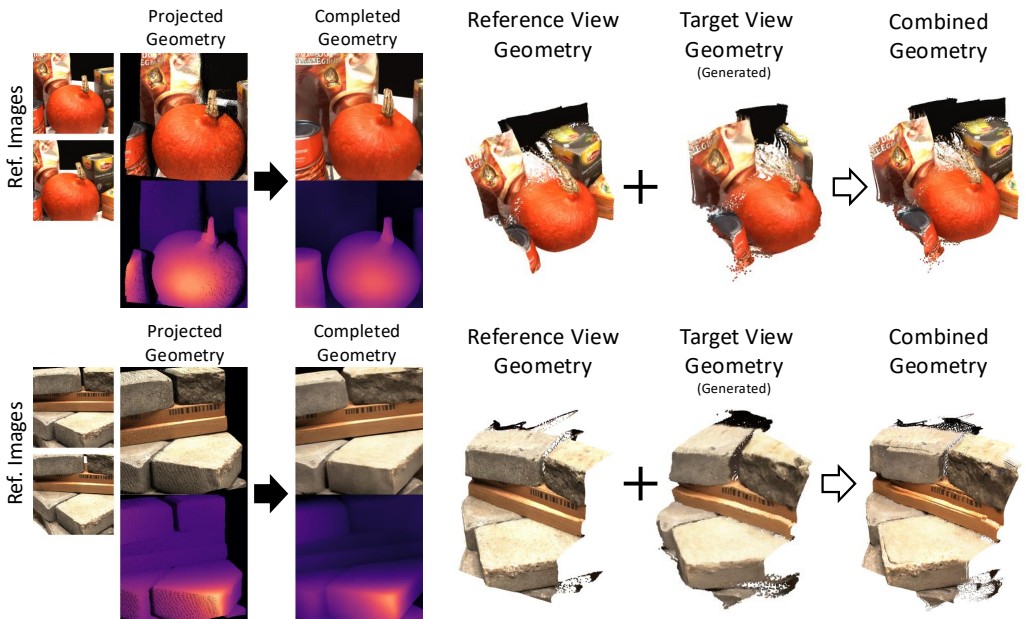

Figure 14: **Alignment visualization with different external model.** We demonstrate alignment visualization between reference view geometry and generated target view geometry, evaluated with DepthAnything V3 (Lin et al., 2025).

## B.5 ALIGNMENT EVALUATION WITH DIFFERENT EXTERNAL POSE PREDICTION MODEL

The alignment of our generated geometry to the predicted geometry is inherently ensured by our architectural design, regardless of where the target camera pose derives from. Specifically, we formulate

geometry generation as inpainting (or completion) of the pointmap that has already been projected to the target camera pose. In this formulation, the model learns to leave the projected geometry unchanged and simply generate extended geometry around it (similar to the image inpainting task), thereby maintaining alignment within the projected geometry according to the given camera pose. Therefore, geometry completion and alignment take place agnostic to the target-camera normalization within our architecture.

To demonstrate this, we provide a cross-predictor validation experiment in Fig. 14, where we employ an alternative external geometry model (DepthAnything v3 (Lin et al., 2025)) at evaluation time to assess alignment between the externally predicted geometry and our model's generated geometry. The visualized results show that our model achieves accurate alignment and high-quality geometry completion regardless of which external prediction model is used for geometry and pose prediction.

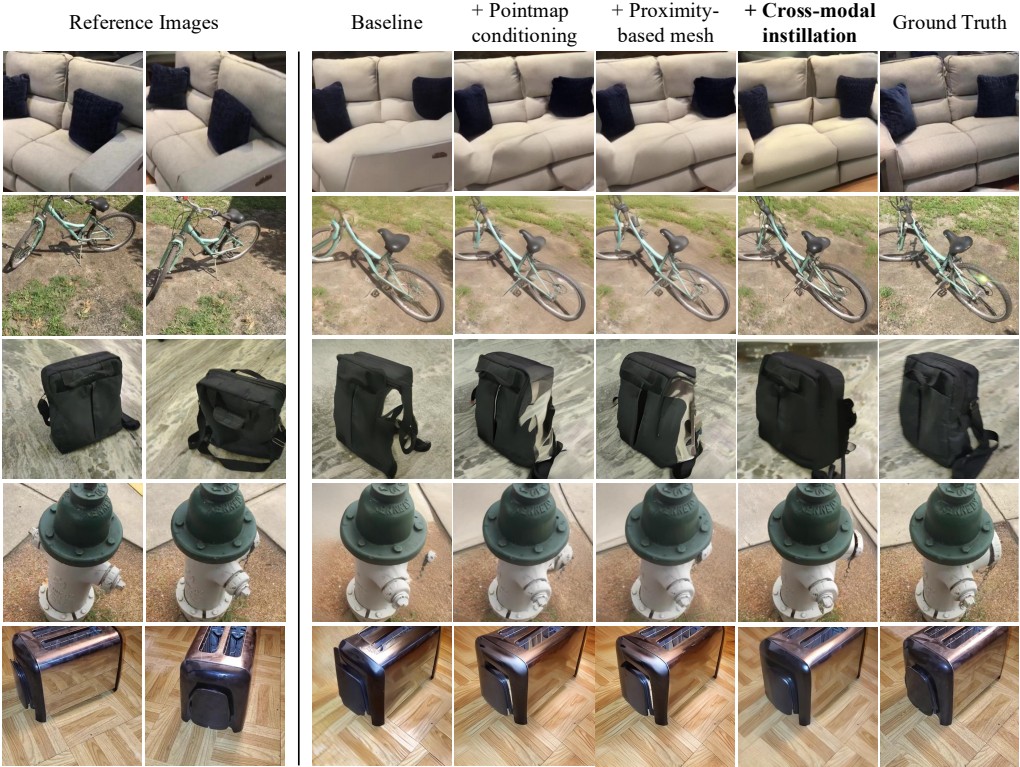

Figure 15: **Ablation results.** Qualitative ablation study across five Co3D scenes demonstrating progressive improvements from naive baseline (spatially incoherent), through pointmap conditioning (improved depth awareness), to mesh-based proximity conditioning (reduced artifacts), and finally cross-modal attention distillation (highest quality with superior consistency). Each component contributes essential capabilities that culminate in state-of-the-art performance with well-aligned modalities and enhanced realism..

## B.6 QUALITATIVE ABLATION

We conduct qualitative ablation experiments across five Co3D scenes in Fig. 15, evaluating four progressive configurations to demonstrate each component's contribution. The totally naive baseline without geometric conditioning struggles with spatial coherence and geometric consistency, producing misaligned features and implausible geometry. Adding pointmap conditioning improves geometric awareness and depth relationships but remains insufficient for complex occlusions and fine-grained details. Mesh-based proximity conditioning yields substantial improvements by providing richer geometric cues, enabling more accurate warping and cleaner geometry synthesis while reducing artifacts. Finally, incorporating cross-modal attention distillation produces the highest quality results through synergistic image-geometry interaction, ensuring well-aligned modalities with superior consistency

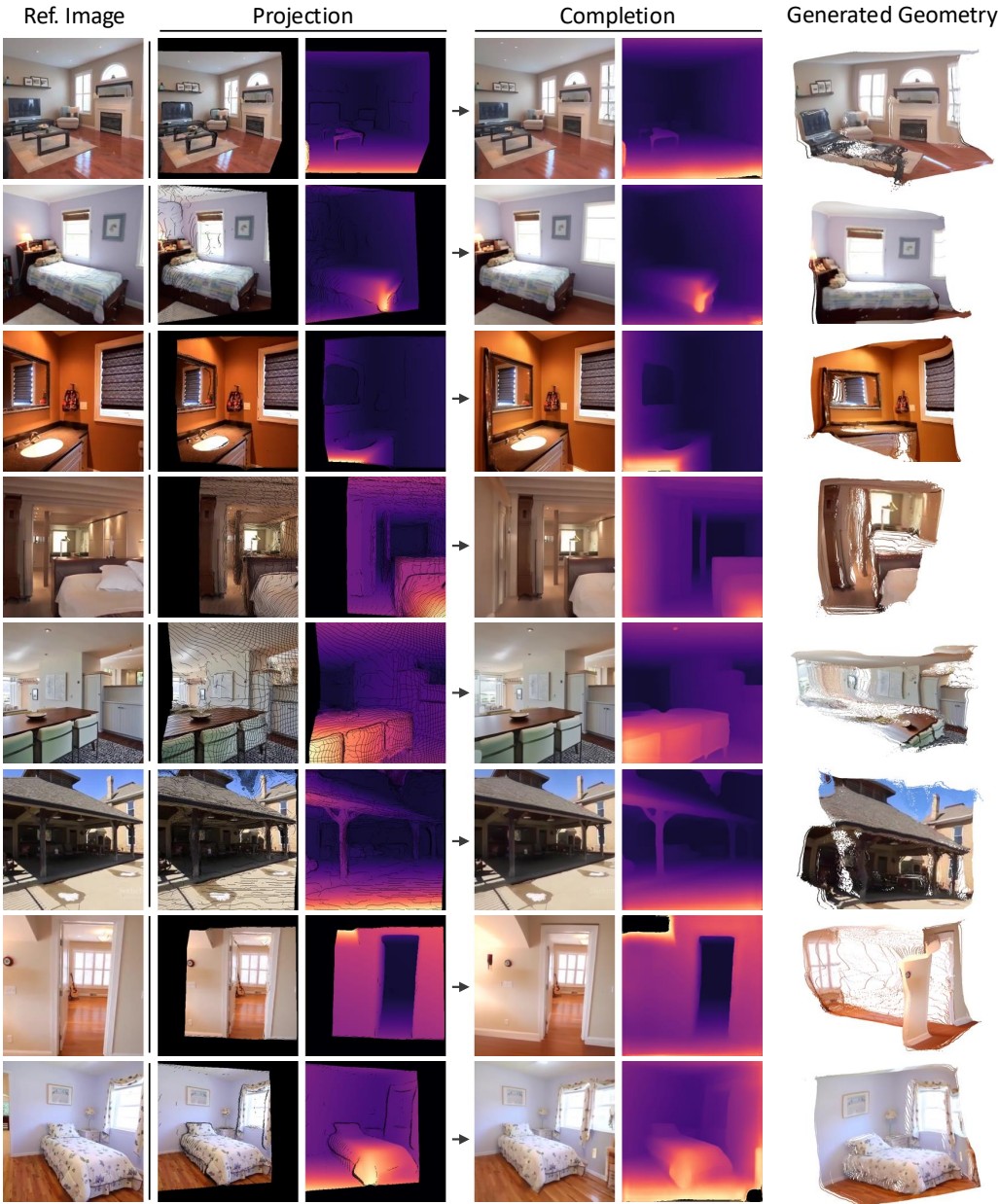

Figure 16: **Qualitative results on single-view extrapolative setting.** Our method can generate coherent novel view images and geometry from a single unposed reference image at extrapolative camera viewpoints, inpainting locations whose information was not given in reference images, while faithfully reconstructing the known regions.

and enhanced realism. This progressive enhancement clearly demonstrates how each component contributes essential capabilities that culminate in our method's state-of-the-art performance.

## B.7 ADDITIONAL SCENE COMPLETION RESULTS

Our proposed method demonstrates remarkable flexibility and scalability across varying numbers of input viewpoints in Fig. 16, showcasing its robust generalization capabilities beyond the two-view training configuration. In single-view novel view synthesis experiments conducted on the RealEstate10K dataset, our model successfully generates high-quality novel views from a single reference image, effectively leveraging the geometric priors learned during training to infer plausible

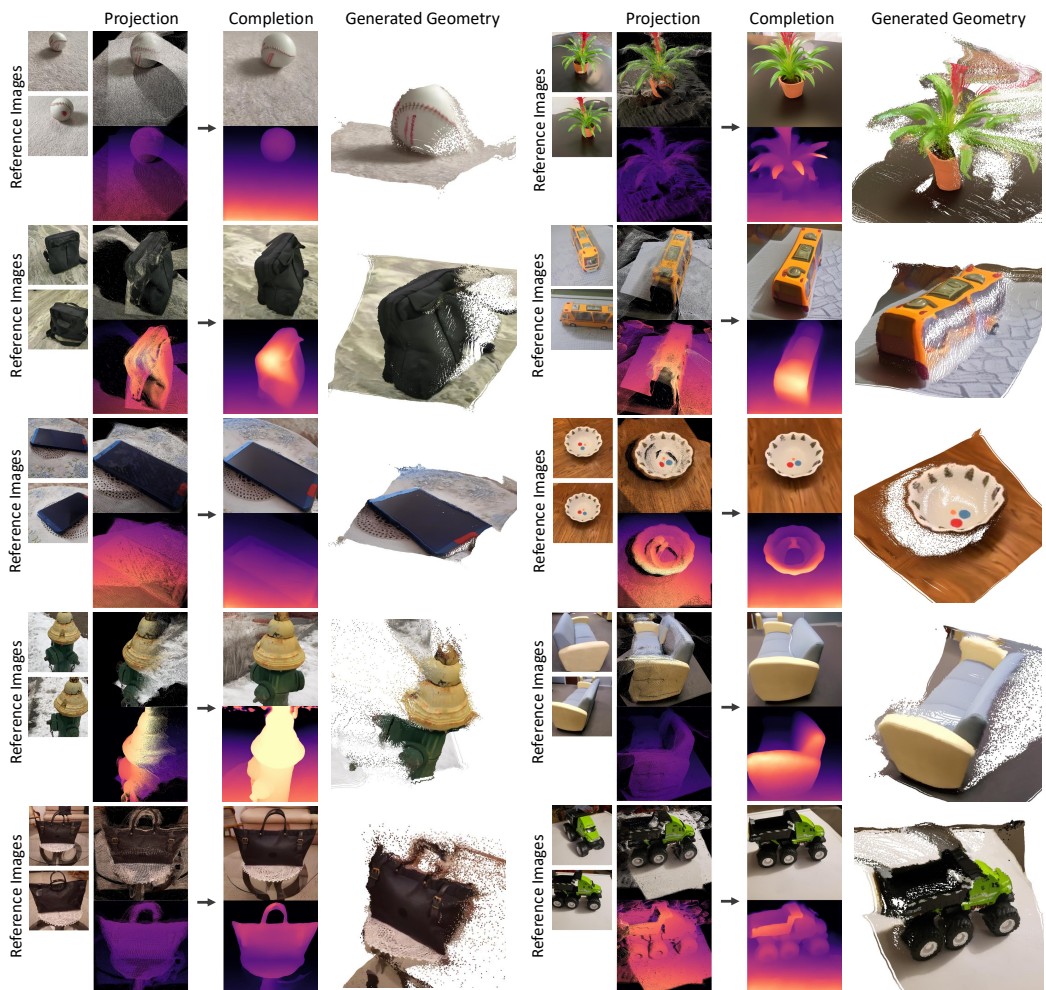

Figure 17: **Qualitative comparison on two-view extrapolative setting.** Our method can generate coherent novel view images and geometry from two unposed reference images at extrapolative camera viewpoints, inpainting locations not seen in reference images, while faithfully reconstructing the known regions.

scene structure and appearance for unseen viewpoints. This single-view capability is particularly challenging as it requires the model to hallucinate significant portions of the target view while maintaining geometric consistency with the limited reference information.

For two-view novel view synthesis, as demonstrated in Fig. 17, we conduct comprehensive evaluations across Co3D Reizenstein et al. (2021) datasets, demonstrating consistent performance improvements when additional reference information becomes available. The model effectively aggregates information from both reference views through our multi-view attention mechanism, resulting in more accurate geometry estimation and higher-fidelity image synthesis. Our architecture's ability to seamlessly handle two-view inputs during inference, despite being trained on this configuration, validates the effectiveness of our correspondence conditioning and attention aggregation strategies.

### B.8 ANALYSIS ON THE NUMBER OF INPUT VIEWPOINTS

Our model conducts aggregated attention to generate novel views from reference images, it can receive an arbitrary number of input viewpoints for generation, as an additional reference image corresponds to simply concatenating additional reference viewpoint's features within our attention architecture. To demonstrate this, in Figure 18, we increase the number of reference viewpoints for a model trained at 2-viewpoint setting and analyze its effects in both image quality and geometric accuracy: the results demonstrate even without being trained on the given number of inputs, our model benefits strongly from additional viewpoints, showing the generalization capability of our aggregated attention architecture to various number of input reference viewpoints.

Table 8: **Quantitative analysis regarding number of viewpoints**. Our quantitative analysis conducted on Co3D dataset shows that our model's multi-view aggregated attention enables it to generalize to an arbitrary number of reference images, with performance consistently increasing with additional reference images.

| Num. of viewpoints | 1 | 2 | 3 | 4 | 5 | 6 | 7 |
|---|---|---|---|---|---|---|---|
| PSNR | 14.62 | 16.43 | 18.65 | 19.17 | 22.90 | 23.11 | 23.45 |

1 View · 2 Views · 3 Views · 4 Views · 5 Views · 6 Views · 7 Views · Ground Truth

Figure 18: **Analysis on number of reference viewpoints.** Our model's multi-view aggregated attention enables it to generalize to an arbitrary number of reference images, with performance consistently increasing with additional reference images.

## C THE USE OF LARGE LANGUAGE MODELS

**Writing assistance.** In this work, large language models (LLMs) were used exclusively for grammatical refinement and polishing of the manuscript's writing. All usages of LLMs were only sentence-level or lower, proofreading for removal of grammatical errors, or rephrasing of certain phrases to enhance the conciseness or academic flair of the sentences originally written by the authors. All technical content, research ideas, experimental design, analysis, and scientific conclusions were developed independently by the authors without LLM assistance. The LLMs did not contribute to research ideation, methodology development, or interpretation of results. We take full responsibility for all content in this paper, including any LLM-assisted grammatical improvements.

