# OpenReview forum: "Aligned Novel View Image and Geometry Synthesis via Cross-modal Attention Instillation"
_ICLR.cc/2026/Conference — ICLR 2026 Poster_

### Official Review · Reviewer_GgBm · 2025-10-27

**Soundness:** 3
**Presentation:** 3
**Contribution:** 2
**Rating:** 6
**Confidence:** 3

**Summary:**

This paper looks at how to perform improved view synthesis, especially in the case of more extreme viewpoint changes where older methods would usually break down (due to slightly incorrect geometry or large reasons for outpainting).

The authors solution is to leverage a predicted 3D point cloud and image generation methodology together in order to improve on the quality of either in isolation by use of a x-attention map that leverages information from both modalities.

The authors propose a pipeline, leveraging prior work (e.g. for point cloud extraction and viewpoint prediction) in order to achieve this.

They achieve good results in comparison to prior work.

**Strengths:**

1. The paper is clearly written and easy enough to follow.

2. The idea is reasonable and makes sense -- to enforce consistency between predicted geometry and viewpoint synthesis. They also propose to use 3D techniques (e.g. conversion to a mesh w/ corresponding normals and depth to allow the model to determine which points are more reliable.

3. The authors evaluate both in/out of distribution and include ablations and standard baselines.

**Weaknesses:**

1. Some of the described intuition is unclear to me: "The image denoising U-Net receives deterministic training signals from the geometry completion network," --> This implies to me that the image-inpainting and geometry prediction models are trained separately but from reading the paper, I don't think that they are. I'm also unsure as to what losses are used to train the geometry prediction model and what it is expected to look like -- my understanding is the loss is on the depmap which the point cloud is projected to. But this is unclear to me from reading the paper.

2. Ablations on the geometry results.
The paper has an ablation in Table 3 on the impact of the geometry losses + cross model instillation for the view synthesis but not on how those same choices impact the geometric results in Table 4.

3. Further comparisons to SOTA
The authors could have included results / comparisons (in terms of speed / performance) to the methods that they say they are improving against by not having to do a separate NERF optimization (e.g. CAT3D). In that case, it would be imp. to compare the performance by considering the need of the authors' work to generate the 3D point cloud in the first place using the other method.

**Questions:**

Please see above.

---

> ### Author Response · Authors · 2025-11-27
> **Response to Reviewer GgBm**
>
> We thank the reviewer for their valuable feedback. We address the questions and comments of the reviewer below.
>
> &nbsp;
>
> ### **Clarifications regarding the geometry branch**
>
> I apologize for lack of clarity in the original manuscript, and I promise to revise it to make it clearer in our paper. To clarify:
>
> - The geometry completion U-Net predicts a pointmap for the novel viewpoint, where each pixel encodes 3D coordinates (resulting in 3 channels). This predicted pointmap is supervised using L2 loss against the ground truth pointmap obtained from the geometry estimation model. To account for scale variations across different scenes, we normalize both ground truth and predicted pointmaps using per-scene statistics (mean and standard deviation), ensuring consistent training dynamics regardless of absolute scene scale.
> - The image denoising U-Net and geometry completion U-Net are jointly trained but remain separate networks that receive distinct loss signals, as described above. Crucially, the geometry loss beneficially affects the image generation branch through backpropagation via cross-modal attention instillation. This architectural coupling allows geometric supervision to improve image generation quality in a geometrically consistent manner.
>
> &nbsp;
>
> ### **Ablation on the geometry results**
>
> In response to your comment, we have conducted an ablation comparing the geometry network with and without cross-modal attention instillation as follows, in Table 1:
>
> **Table 1. Geometry network ablation experiment**
> | Method | **Geometry** |  | **Geometry (Recon)** |  | **Geometry (Inpainting)** |  |
> | --- | --- | --- | --- | --- | --- | --- |
> |  | Abs.Rel↓ | δ₁.₂₅↑ | Abs.Rel↓ &emsp;   &emsp;     | δ₁.₂₅↑ | Abs.Rel↓ | δ₁.₂₅↑ |
> | Naive geometry network | 0.425 | 0.588  &emsp;   &emsp;    | 0.262 | 0.632 &emsp;    &emsp;    | 0.679 | 0.497 |
> | **+ Cross-modal attention instillation** | **0.196** | **0.715** | **0.152** | **0.819** | **0.308** | **0.531** |
>
> Results demonstrate that the incorporation of cross-modal attention instillation (MoAI) significantly improves geometric prediction quality, confirming that the mechanism benefits both modalities. Regarding the full ablation of geometry losses: our current trained checkpoints focus primarily on two configurations—naive training and full instillation—as our paper's main objective was demonstrating how geometry supervision improves image generation. A complete ablation examining individual geometry loss components and their impact on geometric results requires training additional checkpoints, which exceeded our rebuttal timeline. We commit to including this comprehensive geometry ablation in the camera-ready version of our paper. We appreciate your feedback in helping us provide a more thorough analysis.

---

> ### Author Response · Authors · 2025-12-03
> **Response to Reviewer GgBm**
>
> ### **Further comparisons to the state-of-the-art**
>
> We provide additional results demonstrating that our model is competitive with recent large-scale baselines, including the diffusion-based large-scale baselines ViewCrafter [1] and ZeroNVS [2] (following Reviewer wJZj's suggestion), and the large-scale transformer-based model LVSM [3]. First, we emphasize that our model's key strength lies in its ability to directly generate images and geometry at arbitrary camera poses, including extrapolative, far-away viewpoints, in a feedforward manner. Under this setting, our model outperforms other recent state-of-the-art baselines with superior quality and inference times that are orders of magnitude faster.
>
> Figure 1 and Figure 2 of our webpage (https://e4rks32.github.io/supplementary-webpage/#fig1) present qualitative comparisons on in-domain RealEstate10k [4] and in-the-wild Navi [5] datasets, respectively, both under a single-view setting. The results demonstrate our model's superior performance at extrapolative target camera views:
>
> - **LVSM** is unable to handle regions not visible in the reference image, which is the case in such extreme extrapolative viewpoints. This limitation, largely attributable to its lack of generative capability, results in severely degraded imagery.
> - **ZeroNVS** produces plausible novel views and geometry but lacks fidelity. Its camera parameterization requires manual specification of field-of-view, elevation, and content scale for each specific scene, with incorrect values leading to convergence failure, as displayed in the figure. Additionally, ZeroNVS requires approximately 2+ hours per scene for NeRF distillation via Score Distillation Sampling (SDS), making it computationally expensive for practical deployment.
> - **ViewCrafter**: We evaluate both 16-frame and 25-frame models, generating intermediate frames by interpolating camera positions between the reference and target views. Under this extrapolative camera setting, both models produce geometrically degraded imagery and artifacts, as seen in the figures. Furthermore, ViewCrafter requires significantly longer inference time—averaging 209.19 seconds for 25 frames on an A6000 GPU—which is an order of magnitude slower than our model.
> - **MoAI (Ours)** directly generates both novel view images and geometry at arbitrary camera viewpoints in 9.67 seconds on average on an A6000 GPU. Our model produces more plausible and accurate geometry than in comparison to other baseline methods, which can be attributed to cross-modal attention instillation that trains the image generation network toward geometry-aware image generation.
>
> These comparisons demonstrate that our model achieves superior quality and efficiency compared to recent large-scale diffusion baselines, particularly for **challenging extrapolative viewpoints.**
>
> **Quantitative Evaluation:** We further validate our model on DTU dataset under extrapolative view settings (Table 2). We compare against ViewCrafter and LVSM in single-view and two-view settings; ZeroNVS is excluded due to its 1.5-hour per-scene generation time. Our model achieves superior performance across all metrics: +1.52 dB PSNR over ViewCrafter (single-view) and +0.32 dB PSNR over LVSM (two-view), confirming competitive performance on standard benchmarks.
>
> **Table 2. Quantitative evaluation against large-scale baselines**
> |  | View Setting | PSNR | SSIM | LPIPS |
> | --- | --- | --- | --- | --- |
> | ViewCrafter | 1-view | 14.04 | 0.390 | 0.332 |
> | Ours (Single view) | 1-view | 15.56 | 0.609 | 0.184 |
> | LVSM  | 2-view | 15.23 | 0.499 | 0.415 |
> | **Ours (Stereo view)** | **2-view** | **15.58** | **0.615** | **0.184** |
>
> &nbsp;
>
> ### **References**
>
> [1] ViewCrafter: Taming Video Diffusion Models for High-fidelity Novel View Synthesis, Yu et al., TPAMI 2025
>
> [2] ZeroNVS: Zero-Shot 360-Degree View Synthesis from a Single Image, Sargent et al., CVPR 2024
>
> [3] LVSM: A Large View Synthesis Model with Minimal 3D Inductive Bias, Jin et al., ICLR 2025
>
> [4] Stereo Magnification: Learning view synthesis using multiplane images, Zhou et al., SIGGRAPH 2018

---

### Official Review · Reviewer_wJZj · 2025-11-01

**Soundness:** 2
**Presentation:** 3
**Contribution:** 2
**Rating:** 4
**Confidence:** 4

**Summary:**

This paper proposes a diffusion-based inpainting framework to enhance novel view synthesis (NVS) quality, leveraging the 3D foundation model as geometric prior. Specifically, the method utilizes the off-the-shelf geometry predictors and fills both the geometry and image by framing NVS as a completion problem. To ensure alignment between the generated image and geometry, it presents MoAI where the attention maps from the diffusion branch are injected into a parallel geometry diffusion branch. The experiments validate the effectiveness of the proposed approach on several datasets.

**Strengths:**

* The paper is well structured and easy to follow.
* Experiments on Co3D and DTU datasets demonstrate the effectiveness of the proposed method.
* The cross-modal attention installation is interesting to me and has been shown to be effective via the ablation study.

**Weaknesses:**

* The integration of diffusion priors to help NVS has been extensively explored in recent works, like diffusion-aided NeRF/3DGS  reconstruction (Deceptive-NeRF/3DGS [Liu et al.]) and unified frameworks such as ReconX, ViewCrafter, and ZeroNVS. Moreover, the proximity with mesh conditioning is quite standard.  Although the MoAI is interesting, it cannot support the whole paper for a top conference.
* The paper compares primarily against classical or earlier NVS methods (e.g., NeRF variants), but omits several strong diffusion-based or geometry-conditioned NVS works, like ZeroNVS, ViewCrafter, Rand econX.
* Only the quantitative comparison with LVSM is also missing in the paper, which makes it difficult to assess the actual progress relative to the state of the art.
* The runtime and memory consumption are missing.
* The mesh generation with incomplete and noisy points is not easy itself. The robustness of the method is not well evaluated, especially considering that it adopts a rather classical mesh generation.

**Questions:**

Please refer to the weaknesses in the weaknesses part. I am open to being persuaded based on the feedback from the authors.

---

> ### Author Response · Authors · 2025-11-27
> **Response to Reviewer wJZj**
>
> We thank the reviewer for their valuable feedback. We address the questions and comments of the reviewer below.
>
> ### **Regarding the novelty & contribution of our method**
>
> Thank you for your comment. We believe our central contributions — **cross-modal attention instillation** and geometry-based correspondence conditioning — address a fundamental challenge in diffusion-based NVS: achieving geometric consistency without per-scene optimization. We believe our method achieves this in a novel and technically elegant (Reviewer zYBJ) and creative (Reviewer EvEn) manner, with important implications for the research community.
>
> **The key problem:** While recent diffusion-aided NVS methods (CAT3D [1], ReconX [2], ViewCrafter [3]) leverage diffusion priors, they often struggle with ensuring geometric consistency between novel view images. Existing solutions either (1) use explicit 3D representations requiring expensive per-scene optimization (minutes to hours, such as ZeroNVS [4], Reconfusion [5]), or (2) sacrifice geometric fidelity for speed - as we have discussed in detail in our Related Work section.
>
> **Our solution takes a different approach:** We achieve geometric consistency through architectural design rather than optimization. Specifically:
>
> 1. **Correspondence conditioning** uses sparse 3D geometry as a conditioning signal, providing explicit geometric constraints while preserving the feedforward generative capability of diffusion models.
> 2. **Cross-modal attention instillation** enables simultaneous generation of perfectly aligned images and geometry at novel viewpoints without any test-time optimization or 3D representation optimization.
> 3. **Benefits from training with cross-modal instillation**: Our analysis reveals that geometry supervision propagates to the image generation branch through shared attention, as shown in Figure 3 of our main paper, guiding the image generation model to be more geometrically consistent. This implies that the geometry training loss applied at the geometry branch, despite being a training signal for a totally different modality, can positively influence the novel view image generation toward geometric coherence through attention instillation. We believe this is a novel finding with implications beyond our specific architecture.
>
> As we demonstrate below, our approach achieves competitive or superior performance to state-of-the-art methods across multiple domains while maintaining drastically faster inference (9.67s vs. 120s+ for ViewCrafter) while also obtaining scene geometry. We believe this principled approach to geometry-aware generation without optimization opens new directions for diffusion-based 3D reconstruction for the research community in general.

---

> ### Author Response · Authors · 2025-11-27
> **Response to Reviewer wJZj**
>
> ### **Comparison against recent large-scale diffusion-based NVS methods**
>
> We provide additional results demonstrating that our model is competitive with recent large-scale baselines, including the diffusion-based large-scale baselines ViewCrafter [3] and ZeroNVS [4] (the code for ReconX has not been publicly released), and the large-scale transformer-based model LVSM [6]. First, we emphasize that our model's key strength lies in its ability to directly generate images and geometry at arbitrary camera poses, including extrapolative, far-away viewpoints, in a feedforward manner. Under this setting, our model outperforms other recent state-of-the-art baselines with superior quality and inference times that are orders of magnitude faster.
>
> Figure 1 and Figure 2 of our supplementary webpage (https://e4rks32.github.io/supplementary-webpage/#fig1) present qualitative comparisons on in-domain RealEstate10k [7] and in-the-wild Navi [8] datasets, respectively, both under a single-view setting. The results demonstrate our model's superior performance at extrapolative target camera views:
>
> - **LVSM** is unable to handle regions not visible in the reference image, which is the case in such extreme extrapolative viewpoints. This limitation, largely attributable to its lack of generative capability, results in severely degraded imagery.
> - **ZeroNVS** produces plausible novel views and geometry but lacks fidelity. Its camera parameterization requires manual specification of field-of-view, elevation, and content scale for each specific scene, with incorrect values leading to convergence failure, as displayed in the figure. Additionally, ZeroNVS requires approximately 2+ hours per scene for NeRF distillation via Score Distillation Sampling (SDS), making it computationally expensive for practical deployment.
> - **ViewCrafter**: We evaluate both 16-frame and 25-frame models, generating intermediate frames by interpolating camera positions between the reference and target views. Under this extrapolative camera setting, both models produce geometrically degraded imagery and artifacts, as seen in the figures. Furthermore, ViewCrafter requires significantly longer inference time—averaging 209.19 seconds for 25 frames on an A6000 GPU—which is an order of magnitude slower than our model.
> - **MoAI (Ours)** directly generates both novel view images and geometry at arbitrary camera viewpoints in 9.67 seconds on average on an A6000 GPU. Our model produces more plausible and accurate geometry than in comparison to other baseline methods, which can be attributed to cross-modal attention instillation that trains the image generation network toward geometry-aware image generation.
>
> These comparisons demonstrate that our model achieves superior quality and efficiency compared to recent large-scale diffusion baselines, particularly for **challenging extrapolative viewpoints.**
>
> &nbsp;
>
> ### **Quantitative evaluation against LVSM ( + ViewCrafter)**
>
> In continuation to above experiment, we provide quantitative comparisons against LVSM and ViewCrafter in Table 1 below, evaluated at DTU dataset under extrapolative view settings. We compare against ViewCrafter and LVSM in single-view and two-view settings, respectively; ZeroNVS is excluded due to its 1.5-hour per-scene generation time. Our model achieves superior performance across all metrics: +1.52 dB PSNR over ViewCrafter (single-view) and +0.32 dB PSNR over LVSM (two-view), confirming competitive performance on standard benchmarks.
>
> **Table 1. Quantitative evaluation against LVSM and ViewCrafter**
> |  | View Setting | PSNR | SSIM | LPIPS |
> | --- | --- | --- | --- | --- |
> | ViewCrafter | 1-view | 14.04 | 0.390 | 0.332 |
> | Ours (Single-view) | 1-view | 15.56 | 0.609 | 0.184 |
> | LVSM  | 2-view | 15.23 | 0.499 | 0.415 |
> | **Ours (Two-view)** | **2-view** | **15.58** | **0.615** | **0.184** |
>
> &nbsp;
>
> ### **Runtime and memory consumption**
>
> Thank you for pointing this out. Despite employing a dual-branch architecture and multi-view attention mechanisms, our model maintains competitive inference speed and memory efficiency. For 2-view generation with geometry, our model requires an average of 9.81 seconds on an A6000 GPU with 28GB of memory consumption. We emphasize that cross-modal attention instillation plays a crucial role in reducing memory overhead: the geometry network reuses pre-computed attention maps from the image network rather than computing its own, thereby saving both memory and computation. Additionally, our image network can operate standalone for image-only novel view synthesis without the geometry branch, reducing computation requirements to 14GB of memory and 4.3 seconds of inference time —comparable to standard diffusion models. We have included this information in the revision of our paper and thank the reviewer for this constructive suggestion.

---

> ### Author Response · Authors · 2025-11-27
> **Response to Reviewer wJZj**
>
> ### **Robustness evaluation**
>
> Following your comment, we provide additional results demonstrating our model’s strong robustness toward errors and artifacts from external geometry predictors. While it is true that our model relies on the geometry predicted by external models, we find that the generative capability of our diffusion-based model strongly prevents such errors from propagating and degrading the final output. We provide additional experiments as follows:
>
> - In Table 2 below and Figure 4 of our webpage (https://e4rks32.github.io/supplementary-webpage/#fig4), we present both quantitative and qualitative experiments, respectively, examining robustness to geometric noise in the predicted geometry. We apply varying levels of Gaussian noise (standard deviations of 3%, 6%, 10%, and 15% relative to the standard deviation of reference point cloud coordinates) to the point locations in the predicted pointmaps and use these noisy pointmaps as correspondence conditions. Despite high noise levels, our model exhibits minimal performance degradation, demonstrating strong robustness against errors and artifacts in the predicted geometry.
> - In Table 3 below and Figure 5 of our webpage (https://e4rks32.github.io/supplementary-webpage/#fig5), we provide similar experiments regarding the sparsity of the predicted geometry: we randomly mask out a certain percentage of points (30%, 50% and 80%) from the predicted pointmap, and use this sparse pointmap as the correspondence condition. Despite high sparsity in the correspondence condition, our model shows minimal performance degradation even at 80% masking scenario, demonstrating robustness against sparsity in predicted geometry.
> These results confirm that our framework's generative approach effectively mitigates errors from external priors, maintaining our pose-free methodology without compromising output quality.
>
> **Table 2. Quantitative evaluation regarding robustness to geometric noise within the predicted geometry**
> | Condition                     | PSNR      | SSIM      | LPIPS     |
> |-------------------------------|-----------|-----------|-----------|
> | No noise                      | 15.580   &emsp; | 0.615     &emsp;| 0.184   &emsp;  |
> | Gaussian noise (σ = 3%)   &emsp;    | 14.778    | 0.520     | 0.213     |
> | Gaussian noise (σ = 6%)     &emsp;  | 14.501    | 0.507     | 0.225     |
> | Gaussian noise (σ = 10%)   &emsp;   | 14.129    | 0.487     | 0.239     |
> | Gaussian noise (σ = 15%)   &emsp;   | 13.726    | 0.465     | 0.262     |
>
> **Table 3. Quantitative evaluation regarding robustness to increased geometric sparsity in the predicted geometry**
> | Condition                     | PSNR      | SSIM      | LPIPS     |
> |-------------------------------|-----------|-----------|-----------|
> | No masking  | 15.580    | 0.615     | 0.184     |
> | 30% masking    &emsp;  &emsp; &emsp; &emsp;  &emsp;         | 14.683   &emsp; | 0.577   &emsp;  | 0.223  &emsp;   |
> | 50% masking  | 13.610    | 0.468     | 0.272     |
> | 80% masking  | 13.000    | 0.436     | 0.317     |
>
> &nbsp;
>
> ### **Alternative geometry backbone experiment**
>
> As an extension of the robustness experiment, in Table 4 below, we provide an additional experiment with an alternative geometry backbone (DepthAnything V3 [9]), demonstrating our model's compatibility across different external geometry predictors. Specifically, we integrate these alternative backbones into our model (trained with VGGT) without any fine-tuning and evaluate the results on the DTU dataset. Comparisons show that the final generation outputs remain qualitatively similar across the different backbones, confirming our model's robustness to variations in predicted point clouds, as demonstrated in the noise robustness experiments above. This indicates that our model can consistently benefit from advancements in geometry prediction, as improved geometric priors can be seamlessly integrated without requiring architectural modifications or retraining.
>
> **Table 4. Quantitative evaluation regarding alternative geometry backbone**
>
> |  | PSNR | SSIM | LPIPS |
> | --- | --- | --- | --- |
> | w/ VGGT Backbone &emsp;  &emsp; &emsp; &emsp;| 15.580  &emsp; | 0.615  &emsp; | 0.184 &emsp; |
> | w/ DepthAnything V3 Backbone | 15.568 | 0.537 | 0.182 |
>
> &nbsp;
>
> ### **References**
> [1] CAT3D: Create Anything in 3D with Multi-View Diffusion Models, Gao et al., arXiv 2024
>
> [2] ReconX: Reconstruct Any Scene from Sparse Views with Video Diffusion Model, Liu et al., arXiv 2024
>
> [3] ViewCrafter: Taming Video Diffusion Models for High-fidelity Novel View Synthesis, Yu et al., TPAMI 2025
>
> [4] ZeroNVS: Zero-Shot 360-Degree View Synthesis from a Single Image, Sargent et al., CVPR 2024
>
> [5] ReconFusion: 3D Reconstruction with Diffusion Priors, Wu et al., CVPR 2024
>
> [6] LVSM: A Large View Synthesis Model with Minimal 3D Inductive Bias, Jin et al., ICLR 2025
>
> [7] Stereo Magnification: Learning view synthesis using multiplane images, Zhou et al., SIGGRAPH 2018

---

### Official Review · Reviewer_EvEn · 2025-11-02

**Soundness:** 3
**Presentation:** 3
**Contribution:** 3
**Rating:** 6
**Confidence:** 4

**Summary:**

The paper presents a diffusion-based warping-and-inpainting framework that jointly synthesizes novel-view images and geometry from unposed references. It uses off-the-shelf pose/pointmap predictors to project partial 3D into the target view, then completes both modalities via conditional denoising. The core idea, cross-Modal Attention Instillation (MoAI), replaces the geometry U-Net’s spatial attention with that of the image U-Net during training and inference to enforce image–geometry alignment and leverage semantic correspondences. To handle noisy projections, proximity-based mesh conditioning converts sparse point clouds to meshes, augments conditions with depth/normal cues, and masks implausible surfaces. Experiments on DTU and RealEstate10K show strong extrapolative NVS, competitive interpolation, aligned pointmaps enabling 3D completion, and consistent gains from pointmap/mesh conditioning and MoAI.

**Strengths:**

- Joint diffusion for images and geometry with cross-modal attention instillation (MoAI) is a simple, original coupling; using image attention to guide geometry and vice versa is creative, and proximity-based mesh conditioning is a practical improvement over raw pointmaps.

- Empirical results are strong in extrapolative settings on DTU and RealEstate10K, with clear additive gains in ablations (pointmap → mesh → MoAI) and the ability to benefit from more input views at test time despite two-view training.

- The method is clearly described with equations and rationale; the motivation for MoAI is well illustrated, and training/inference details are transparent enough to reproduce.

- The work is significant for pose-free extrapolative NVS and image–geometry alignment, producing aligned outputs that enable 3D completion and benefit downstream reconstruction and content creation.

**Weaknesses:**

- Reliance on off-the-shelf geometry/pose predictors
The pipeline depends on VGGT/MASt3R-style predictors for both training supervision (pseudo GT) and inference conditioning. This creates a ceiling tied to those models’ biases and errors, and complicates fairness: improvements may partially reflect better use of VGGT rather than intrinsic advances. Please include: (i) a robustness study under degraded pointmaps/poses (noise, sparsity, biased scale), (ii) an alternative backbone (e.g., DUSt3R/MASt3R vs. VGGT) to show generality.

- MoAI ablations are underspecified
The paper replaces geometry U-Net attention with image U-Net attention, but lacks: (i) analysis of which layers/stages matter most, (ii) partial vs. full instillation, and (iii) compute/memory overhead.

- Geometry alignment claims hinge on pseudo GT and camera-space normalization
Depth/pointmap “alignment” is evaluated with pseudo labels derived from the same family of predictors, and benefits from target-camera-space normalization. To avoid confirmation bias, add: (i) cross-predictor validation (e.g., train with VGGT pseudo GT, evaluate with MASt3R/Colmap), and (ii) ablations that remove target-camera normalization to quantify true contribution and residual misalignment.

- Limited analysis of multi-view scaling and consistency across targets
The model can accept more reference views at test time, but consistency across multiple simultaneously generated target views is not evaluated. Add: (i) cross-target consistency metrics (e.g., photometric and geometric cycle consistency), and (ii) performance scaling plots vs. number of references with compute trade-offs.

**Questions:**

- Scope of “pose-free”: The method estimates poses/pointmaps from references via an external model and then conditions the diffusion process. In practical deployments, how are target cameras specified? Are you assuming the user provides an arbitrary camera, or do you sample target poses? If target intrinsics/extrinsics are perturbed, how robust is synthesis?
- MoAI mechanism and design choices: Which attention layers are instilled (early/mid/late; which resolutions)? Did you try partial instillation (e.g., only shallow or only deep blocks) or mixing a fraction of the image U-Net attention instead of full replacement? Please include layer-wise ablations and attention visualizations before/after MoAI on several scenes to clarify what correspondences are transferred and where the largest gains arise.
- Comparisons to more baselines: Could you add AnySplat and a DUSt3R+PnP+splatting pipeline as additional pose-free baselines under 2–3 views, at least on DTU/RE10K subsets? If they fail at larger extrapolation, a controlled comparative failure analysis (with runtime and failure modes) would strengthen your positioning.
- Failure cases and qualitative diagnostics: A short taxonomy of failure modes and their link to upstream geometry errors vs. diffusion inpainting would be very helpful, along with attention visualizations that reveal where MoAI succeeds or fails.

---

> ### Author Response · Authors · 2025-11-27
> **Response to Reviewer EvEn**
>
> &nbsp;
>
> We thank the reviewer for their valuable feedback. We address the questions and comments of the reviewer below.
>
> &nbsp;
>
> ## Regarding Reliance on off-the-shelf geometry/pose predictors
>
> ### **Robustness Experiments**
>
> Following your comment, we provide additional results demonstrating our model’s strong robustness toward errors and artifacts from external geometry predictors. While it is true that our model relies on the geometry predicted by external models, we find that the generative capability of our diffusion-based model strongly prevents such errors from propagating and degrading the final output. We provide additional experiments as follows:
>
> - In Table 1 below and Figure 4 of our supplementary webpage (https://e4rks32.github.io/supplementary-webpage/#fig4), we present both quantitative and qualitative experiments, respectively, examining robustness to geometric noise in the predicted geometry. We apply varying levels of Gaussian noise (standard deviations of 3%, 6%, 10%, and 15% relative to the standard deviation of reference point cloud coordinates) to the point locations in the predicted pointmaps and use these noisy pointmaps as correspondence conditions. Despite high noise levels, our model exhibits minimal performance degradation, demonstrating strong robustness against errors and artifacts in the predicted geometry.
> - In Table 2 below and Figure 5 of our webpage (https://e4rks32.github.io/supplementary-webpage/#fig5), we provide similar experiments regarding the sparsity of the predicted geometry: we randomly mask out a certain percentage of points (30%, 50% and 80%) from the predicted pointmap, and use this sparse pointmap as the correspondence condition. Despite high sparsity in the correspondence condition, our model shows minimal performance degradation even at 80% masking scenario, demonstrating robustness against sparsity in predicted geometry.
> These results confirm that our framework's generative approach effectively mitigates errors from external priors, maintaining our pose-free methodology without compromising output quality.
>
> **Table 1. Quantitative evaluation regarding robustness to geometric noise within the predicted geometry**
> | Condition                     | PSNR      | SSIM      | LPIPS     |
> |-------------------------------|-----------|-----------|-----------|
> | No noise                      | 15.580   &emsp; | 0.615     &emsp;| 0.184   &emsp;  |
> | Gaussian noise (σ = 3%)   &emsp;    | 14.778    | 0.520     | 0.213     |
> | Gaussian noise (σ = 6%)     &emsp;  | 14.501    | 0.507     | 0.225     |
> | Gaussian noise (σ = 10%)   &emsp;   | 14.129    | 0.487     | 0.239     |
> | Gaussian noise (σ = 15%)   &emsp;   | 13.726    | 0.465     | 0.262     |
>
>
> **Table 2. Quantitative evaluation regarding robustness to increased geometric sparsity in the predicted geometry**
> | Condition                     | PSNR      | SSIM      | LPIPS     |
> |-------------------------------|-----------|-----------|-----------|
> | No masking                    | 15.580    | 0.615     | 0.184     |
> | 30% masking    &emsp;  &emsp; &emsp; &emsp;  &emsp;         | 14.683   &emsp; | 0.577   &emsp;  | 0.223  &emsp;   |
> | 50% masking                   | 13.610    | 0.468     | 0.272     |
> | 80% masking                   | 13.000    | 0.436     | 0.317     |
>
> &nbsp;
>
> ### **Alternative geometry backbone experiment**
>
> We provide in Table 3 below an additional experiment with an alternative geometry backbone (DepthAnything V3 [1]), demonstrating our model's compatibility across different external geometry predictors.  Specifically, we integrate these alternative backbones into our model (trained with VGGT) without any fine-tuning and evaluate the results on the DTU dataset. Comparisons show that the final generation outputs remain qualitatively similar across the different backbones, confirming our model's robustness to variations in predicted point clouds, as demonstrated in the noise robustness experiments above. This indicates that our model can consistently benefit from advancements in geometry prediction, as improved geometric priors can be seamlessly integrated without requiring architectural modifications or retraining.
>
> **Table 3. Quantitative evaluation regarding alternative geometry backbone**
> |  | PSNR | SSIM | LPIPS |
> | --- | --- | --- | --- |
> | w/ VGGT Backbone &emsp;  &emsp; &emsp; &emsp;| 15.580  &emsp; | 0.615  &emsp; | 0.184 &emsp; |
> | w/ DepthAnything V3 Backbone | 15.568 | 0.537 | 0.182 |

---

> ### Author Response · Authors · 2025-11-27
> **Response to Reviewer EvEn**
>
> ### **Regarding the ablation experiment of attention instillation layers**
>
> Thank you for this insightful comment. In our current "naive" implementation, we have employed full instillation across all spatial attention layers within both the image and geometry diffusion models.
>
> Following your suggestion, we have conducted layer-wise ablation studies examining which U-Net layers benefit most from cross-modal attention instillation. We trained four configurations from scratch, applying instillation to different layer groups based on spatial dimensions (results in Table 4 below - please note that values are lower than our main results due to abbreviated training time within the rebuttal period). Results show that instillation mid-deep layers [4,5,9,10] achieve optimal performance, as we hypothesize that the intermediate features effectively balance spatial detail and semantic understanding for geometric supervision. In contrast, we observe shallow layers provide minimal benefit, possibly due to insufficient semantic context. The results demonstrate that training by partial instillation, with the exception of the shallow layers scenario, outperforms our naive method of full instillation: we believe this is a substantial finding, and we thank the reviewer for the constructive comment. We will include a detailed analysis of this experiment in the final version of our paper.
>
> **Table 4. Ablation regarding instillation layers**
>
> |  | PSNR | SSIM | LPIPS |
> | --- | --- | --- | --- |
> | Shallow layers [0,1,14,15] | 13.18 &emsp; &emsp;| 0.536&emsp; &emsp;| 0.348&emsp; &emsp;|
> | Mid-shallow layers [2,3,11,12,13] | 13.78 | 0.584 | 0.309 |
> | **Mid-deep layers [4,5,9,10]** | **13.81** | **0.587** | **0.306** |
> | Deep layers [6,7,8] | 13.66 | 0.573 | 0.318 |
> | Naive | 13.56 | 0.562 | 0.327 |
>
> &nbsp;
>
> ### **Compute complexity and practicality**
>
> Thank you for pointing this out. Despite employing a dual-branch architecture and multi-view attention mechanisms, our model maintains competitive inference speed and memory efficiency. For 2-view generation with geometry, our model requires an average of 9.81 seconds on an A6000 GPU with 28GB of memory consumption. We emphasize that cross-modal attention instillation plays a crucial role in reducing memory overhead: the geometry network reuses pre-computed attention maps from the image network rather than computing its own, thereby saving both memory and computation. Additionally, our image network can operate standalone for image-only novel view synthesis without the geometry branch, reducing computation requirements to 14GB of memory and 4.3 seconds of inference time —comparable to standard diffusion models.
>
> &nbsp;
>
> ### **Regarding geometry alignment and camera pose**
>
> **We would like to provide an important clarification regarding "alignment":** The alignment of our generated geometry to the predicted geometry is inherently ensured by our architectural design, regardless of where the target camera pose derives from. Specifically, we formulate geometry generation as inpainting (or completion) of the pointmap that has already been projected to the target camera pose. In this formulation, the model learns to leave the projected geometry unchanged and simply generate extended geometry around it (similar to the image inpainting task), thereby maintaining alignment within the projected geometry according to the given camera pose. Therefore, it can be said that geometry completion and alignment take place agnostic to the target-camera normalization within our architecture.
>
> To demonstrate this, following your suggestion, we provide a cross-predictor validation experiment in Figure 7 of our webpage (https://e4rks32.github.io/supplementary-webpage/#fig7), where we employ an alternative external geometry model (DepthAnything v3) at evaluation time to assess alignment between the externally predicted geometry and our model's generated geometry. The visualized results show that our model achieves accurate alignment and high-quality geometry completion regardless of which external model is used for geometry and pose prediction.
>
> &nbsp;
>
> ### **Additional Baselines Experiments**
>
> Thank you for this valuable suggestion. We agree that comparison with AnySplat [2] and DUSt3R+PnP+splatting pipelines would strengthen our evaluation. Unfortunately, implementing these baselines required substantial engineering effort and computational resources that exceeded our rebuttal timeline. However, we are committed to including these comparisons in the camera-ready version of our paper. We appreciate your constructive feedback in helping us improve the robustness of our work.
>
> &nbsp;
>
> ### **References**
> [1] Depth Anything V3: Recovering the Visual Space from Any Views, Lin et al., arXiv 2025
>
> [2] AnySplat: Feed-forward 3D Gaussian Splatting from Unconstrained Views, Jiang et al., ACM TOG 2025

---

### Official Review · Reviewer_zYBJ · 2025-11-10

**Soundness:** 3
**Presentation:** 2
**Contribution:** 2
**Rating:** 4
**Confidence:** 3

**Summary:**

This paper proposes a diffusion-based framework for few-shot novel view synthesis (NVS) that generates both aligned novel view images and corresponding geometries. Unlike prior approaches (e.g., PixelSplat, MVSplat, or diffusion-based GenWarp) that require dense camera poses or handle only interpolative view synthesis, this work aims to address the pose-free extrapolative NVS setting.

The key innovation is Cross-Modal Attention Instillation (MoAI): a mechanism that transfers attention maps from an image diffusion branch to a geometry diffusion branch. This encourages alignment between synthesized appearance and predicted geometry, mitigating inconsistencies that arise in conventional inpainting-based pipelines.

**Strengths:**

- Novel and technically elegant contribution:
The cross-modal attention instillation mechanism is conceptually clean and effectively bridges the gap between image synthesis and geometry completion. It’s a natural extension of diffusion-based correspondence learning and could influence future cross-domain conditioning designs.

- Comprehensive evaluation:
The experiments cover both extrapolative and interpolative regimes, multiple datasets (Co3D, DTU, RealEstate10K), and ablation analyses (Table 3–5) that clearly isolate contributions of each design component (pointmap conditioning, mesh proximity, MoAI).

- Strong empirical results:
The method consistently outperforms existing approaches across metrics (e.g., PSNR↑17.4 vs 14.3 on DTU extrapolative), demonstrating robust generalization, especially in pose-free and single-view setups.

- Interpretability and coherence:
The qualitative figures (Fig. 3, 6, 9) convincingly show improvements in structural alignment, artifact reduction, and cross-view consistency. The framework maintains geometric realism without explicit NeRF optimization.

- Scalability and flexibility:
The model generalizes to variable numbers of input views (Table 5), highlighting strong architectural modularity and robustness.

**Weaknesses:**

- Limited exploration of generalization across domains and semantics:
It remains unclear whether the approach scales to in-the-wild scenes (e.g., urban/street-level data) or semantic diversity (humans, animals, etc.).

- Dependence on pretrained geometry estimators (e.g., VGGT, Marigold):
The framework relies heavily on the quality of external predictors. Errors in these priors can propagate, which partially undermines the claim of being “pose-free” or “self-contained.”

- Lack of detailed comparison to recent large-scale diffusion baselines:
While comparisons to Zero123, GenWarp, and MVDream are discussed, newer world-consistent 3D diffusion models should be quantitatively evaluated to position the work more clearly.

- Compute complexity and practicality:
Training with dual-branch diffusion networks (image + geometry) and multi-view attention is computationally expensive; inference speed and memory footprint are not reported.

- Missing discussion of failure cases:
The paper does not analyze scenarios where MoAI might fail, e.g., inconsistent shadows or hallucinated surfaces due to semantic-geometry conflicts.

**Questions:**

- How does MoAI compare to simply concatenating geometry and image features before attention?
Could this mechanism be replaced with a more general form of cross-attention sharing?

- What happens if the pretrained geometry network is replaced with a lightweight learned depth prior, does performance drop significantly?

- Does the model maintain physical consistency (e.g., shading or occlusion correctness) when extrapolating beyond 90° view differences?

- Could the method be extended to temporal coherence (e.g., video novel view synthesis)?

---

> ### Author Response · Authors · 2025-11-27
> **Response to Reviewer zYBJ**
>
> We thank the reviewer for their valuable feedback. We address the questions and comments of the reviewer below.
>
> ### **Regarding the exploration of generalization across domains**
>
> We provide additional results demonstrating our model's strong generalization to unseen domains, including in-the-wild and urban data. We conducted experiments on three new datasets from different domains: the CityScapes [1] dataset, containing urban street-view multi-view images; the MegaDepth [2] dataset, containing in-the-wild internet photos of popular landmarks with significant viewpoint and illumination variation; and the Navi [3] dataset, containing in-the-wild multi-view object image collections. None of these datasets was used during training, thereby validating our model's generalization capabilities. Figure 2 and Figure 3 in our supplementary webpage (https://e4rks32.github.io/supplementary-webpage/#fig3) present qualitative results demonstrating high-quality novel view synthesis and geometry reconstruction on in-the-wild data, confirming strong generative performance across diverse domains.
>
> &nbsp;
>
> ### **Regarding dependence on pretrained geometry estimators**
>
> We provide additional results demonstrating our model’s strong robustness toward errors and artifacts from external geometry predictors. While it is true that our model relies on the geometry predicted by external models, we find that the generative capability of our diffusion-based model strongly prevents such errors from propagating and degrading the final output. We provide additional experiments as follows:
>
> - In Table 1 below and Figure 4 of our webpage (https://e4rks32.github.io/supplementary-webpage/#fig4), we present both quantitative and qualitative experiments, respectively, examining robustness to geometric noise in the predicted geometry. We apply varying levels of Gaussian noise (standard deviations of 3%, 6%, 10%, and 15% relative to the standard deviation of reference point cloud coordinates) to the point locations in the predicted pointmaps and use these noisy pointmaps as correspondence conditions. Despite high noise levels, our model exhibits minimal performance degradation, demonstrating strong robustness against errors and artifacts in the predicted geometry.
> - In Table 2 below and Figure 5 of our webpage (https://e4rks32.github.io/supplementary-webpage/#fig5), we provide similar experiments regarding the sparsity of the predicted geometry: we randomly mask out a certain percentage of points (30%, 50% and 80%) from the predicted pointmap, and use this sparse pointmap as the correspondence condition. Despite high sparsity in the correspondence condition, our model shows minimal performance degradation even at 80% masking scenario, demonstrating robustness against sparsity in predicted geometry.
> These results confirm that our framework's generative approach effectively mitigates errors from external priors, maintaining our pose-free methodology without compromising output quality.
>
> **Table 1. Quantitative evaluation regarding robustness to geometric noise within the predicted geometry**
> | Condition | PSNR| SSIM  | LPIPS|
> |-|-|-|-|
> | No noise | 15.580   &emsp; | 0.615 &emsp;| 0.184 &emsp;  |
> | Gaussian noise (σ = 3%)|14.778| 0.520 | 0.213|
> | Gaussian noise (σ = 6%) |14.501| 0.507 | 0.225|
> | Gaussian noise (σ = 10%)|14.129 | 0.487 | 0.239|
> | Gaussian noise (σ = 15%)|13.726 | 0.465 | 0.262|
>
>
> **Table 2. Quantitative evaluation regarding robustness to increased geometric sparsity in the predicted geometry**
> | Condition| PSNR| SSIM | LPIPS|
> |-|-|-|-|
> | No masking| 15.580 | 0.615| 0.184|
> | 30% masking &emsp;  &emsp; &emsp; &emsp;  &emsp;| 14.683 &emsp; | 0.577 &emsp;  | 0.223 &emsp; |
> | 50% masking |13.610|0.468| 0.272|
> | 80% masking |13.000|0.436| 0.317|
> &nbsp;
> ### **Alternative geometry backbone experiment**
> To show our model's robustness to external models, in Table 3 below, we provide an additional experiment with an alternative geometry backbone (DepthAnything V3 [9]), demonstrating our model's compatibility across different external geometry predictors. Specifically, we integrate these alternative backbones into our model (trained with VGGT) **without any training** and evaluate the results on the DTU dataset. Comparisons show that the final generation outputs remain qualitatively similar across the different backbones, confirming our model's robustness to variations in predicted point clouds, as demonstrated in the noise robustness experiments above. This indicates that our model can consistently benefit from advancements in geometry prediction, as improved geometric priors can be seamlessly integrated without requiring architectural modifications or retraining.
>
> **Table 3. Quantitative evaluation regarding alternative geometry backbone**
> |  | PSNR | SSIM | LPIPS |
> | --- | --- | --- | --- |
> | w/ VGGT Backbone &emsp;  &emsp; &emsp; &emsp;| 15.580&emsp; | 0.615&emsp; | 0.184 &emsp; |
> | w/ DepthAnything V3 Backbone | 15.568 | 0.537 | 0.182 |

---

> ### Author Response · Authors · 2025-11-27
> **Response to Reviewer zYBJ**
>
> ### **Comparison against recent large-scale diffusion baselines**
>
> We provide additional results demonstrating that our model is competitive with recent large-scale baselines, including the diffusion-based large-scale baselines ViewCrafter [4] and ZeroNVS [5] (following Reviewer wJZj's suggestion), and the large-scale transformer-based model LVSM [6]. First, we emphasize that our model's key strength lies in **its ability to directly generate images and geometry at arbitrary camera poses**, including extrapolative, far-away viewpoints, in a feedforward manner. Under this setting, our model outperforms other recent state-of-the-art baselines with superior quality and inference times that are orders of magnitude faster.
>
> Figure 1 and Figure 2 of our webpage (https://e4rks32.github.io/supplementary-webpage/#fig1) present qualitative comparisons on in-domain RealEstate10k [7] and in-the-wild Navi [3] datasets, respectively, both under a single-view setting. The results demonstrate our model's superior performance at extrapolative target camera views:
>
> - **LVSM** is unable to handle regions not visible in the reference image, which is the case in such extreme extrapolative viewpoints. This limitation, largely attributable to its lack of generative capability, results in severely degraded imagery.
> - **ZeroNVS** produces plausible novel views and geometry but lacks fidelity. Its camera parameterization requires manual specification of field-of-view, elevation, and content scale for each specific scene, with incorrect values leading to convergence failure, as displayed in the figure. Additionally, ZeroNVS requires approximately 2+ hours per scene for NeRF distillation via Score Distillation Sampling (SDS), making it computationally expensive for practical deployment.
> - **ViewCrafter**: We evaluate both 16-frame and 25-frame models, generating intermediate frames by interpolating camera positions between the reference and target views. Under this extrapolative camera setting, both models produce geometrically degraded imagery and artifacts, as seen in the figures. Furthermore, ViewCrafter requires significantly longer inference time—averaging 209.19 seconds for 25 frames on an A6000 GPU—which is an order of magnitude slower than our model.
> - **MoAI (Ours)** directly generates both novel view images and geometry at arbitrary camera viewpoints in 9.67 seconds on average on an A6000 GPU. Our model produces more plausible and accurate geometry than in comparison to other baseline methods, which can be attributed to cross-modal attention instillation that trains the image generation network toward geometry-aware image generation.
>
> These comparisons demonstrate that our model achieves superior quality and efficiency compared to recent large-scale diffusion baselines, particularly for **challenging extrapolative viewpoints.**
>
> **Quantitative Evaluation:** We further validate our model on DTU dataset under extrapolative view settings (Table 4). We compare against ViewCrafter and LVSM in single-view and two-view settings; ZeroNVS is excluded due to its 1.5-hour per-scene generation time. Our model achieves superior performance across all metrics: +1.52 dB PSNR over ViewCrafter (single-view) and +0.32 dB PSNR over LVSM (two-view), confirming competitive performance on standard benchmarks.
>
> **Table 4. Quantitative evaluation against large-scale baselines**
> |  | View Setting | PSNR | SSIM | LPIPS |
> | --- | --- | --- | --- | --- |
> | ViewCrafter | 1-view | 14.04 | 0.390 | 0.332 |
> | Ours (Single-view) | 1-view | 15.56 | 0.609 | 0.184 |
> | LVSM  | 2-view | 15.23 | 0.499 | 0.415 |
> | **Ours (Two-view)** | **2-view** | **15.58** | **0.615** | **0.184** |
>
> &nbsp;
>
> ### **Comparison to a more general form of cross-attention sharing**
>
> When simply concatenating geometry and image features, in our experiments, we observed that erroneous feature mixing frequently occurs between the two modalities. Specifically, because both image and geometry use the same VAE for encoding and decoding (following the standard procedure in Marigold [8]), geometry features tend to influence image features and vice versa, resulting in degraded performance for both modalities. Our MoAI architecture addresses this issue through separate image and geometry generation branches that share a single attention map. This design enables the sharing of mutually important geometric and correspondence information for alignment and improved generation quality, while architecturally preventing feature mixing. Thank you for this insightful question!

---

> ### Author Response · Authors · 2025-11-27
> **Response to Reviewer zYBJ**
>
> &nbsp;
>
> ### **Compute complexity and practicality**
>
> Thank you for pointing this out. Despite employing a dual-branch architecture and multi-view attention mechanisms, our model maintains competitive inference speed and memory efficiency. For 2-view generation with geometry, our model requires an average of 9.81 seconds on an A6000 GPU with 28GB of memory consumption. We emphasize that cross-modal attention instillation plays a crucial role in reducing memory overhead: the geometry network reuses pre-computed attention maps from the image network rather than computing its own, thereby saving both memory and computation. Additionally, our image network can operate standalone for image-only novel view synthesis without the geometry branch, reducing computation requirements to 14GB of memory and 4.3 seconds of inference time —comparable to standard diffusion models. We have included this information in the revision of our paper and thank the reviewer for this constructive suggestion.
>
> &nbsp;
>
> ### **Replacing the external geometry predictor with a lightweight learned depth prior**
>
> Thank you for your comment. In the single-view setting, as shown in Table 5, replacing the pretrained geometry network (e.g., DUSt3R [9] or VGGT [10]) with a lightweight depth prior (such as a monocular depth estimation model) does not negatively affect generation quality. To demonstrate this, we have conducted an additional experiment using a light version of DepthAnything (v3) (DA3-SMALL model, 34.3M Params) [11] model as our external model under a single reference view scenario (without any additional training), in which the monocular depth predicted by the DepthAnything model is turned into a point cloud, then used as a correspondence condition. The result does show degradation, but it is not a significant drop from the original setting in which the VGGT backbone is used (also single-view setting). We believe this result also demonstrates our model's robustness to using various external geometry prediction models for correspondence conditioning.
>
> **Table 5. Quantitative evaluation using a lightweight depth model as an external predictor**
> |  | PSNR | SSIM | LPIPS |
> | --- | --- | --- | --- |
> | Original setting (w/ VGGT, single-view) &emsp;  &emsp; &emsp; &emsp;| 15.560  &emsp; | 0.609  &emsp; | 0.184 &emsp; |
> | w/ DepthAnything V3-SMALL (single-view)  &emsp;  | 14.554 | 0.472 | 0.241 |
>
> However, for multi-view generation, a critical challenge arises when naively using monocular depth priors: per-view predicted depths have to be aligned with each other, but each prediction exhibits scale-shift ambiguities inherent to monocular depth estimation. This alignment problem degrades generation quality, as the correspondence condition cannot provide meaningful cross-view geometric information to the network. We thank the reviewer for the valuable suggestion.
>
> &nbsp;
>
> ### **Regarding physical consistency during extrapolation**
>
> Thank you for pointing this out. Yes, our model maintains physical consistency when extrapolating beyond 90° view differences, including correct shading and occlusion handling. Figure 4 in our main paper and Figure 8 in our supplementary webpage (https://e4rks32.github.io/supplementary-webpage/#fig8) demonstrate numerous cases where our model generates target views more than 90° away from the reference viewpoints, requiring generation of completely unseen regions (e.g., the reverse side of the toaster in Fig. 4). Our results show that the model successfully completes both image and geometry for these unseen views. As illustrated in Figure 8, shadows and reflections are generated correctly and plausibly, demonstrating the strong generative capability of our diffusion-based NVS approach.
>
> &nbsp;
>
> ### **References**
>
> [1] The Cityscapes Dataset for Semantic Urban Scene Understanding, Cordts et al., CVPR 2016
>
> [2] MegaDepth: Learning Single-View Depth Prediction from Internet Photos, Li and Snavely, CVPR 2018
>
> [3] NAVI: Category-Agnostic Image Collections with High-Quality 3D Shape and Pose Annotations, Jampani et al., NeurIPS 2023
>
> [4] ViewCrafter: Taming Video Diffusion Models for High-fidelity Novel View Synthesis, Yu et al., TPAMI 2025
>
> [5] ZeroNVS: Zero-Shot 360-Degree View Synthesis from a Single Image, Sargent et al., CVPR 2024
>
> [6] LVSM: A Large View Synthesis Model with Minimal 3D Inductive Bias, Jin et al., ICLR 2025
>
> [7] Stereo Magnification: Learning view synthesis using multiplane images, Zhou et al., SIGGRAPH 2018
>
> [8] Repurposing Diffusion-Based Image Generators for Monocular Depth Estimation, Ke et al., CVPR 2024
>
> [9] DUSt3R: Geometric 3D Vision Made Easy, Wang et al., CVPR 2024
>
> [10] VGGT: Visual Geometry Grounded Transformer, Wang et al., CVPR 2025
>
> [11] Depth Anything V3: Recovering the Visual Space from Any Views, Lin et al., arXiv 2025

---

### Author Response · Authors · 2025-11-29
**General Comment**

We thank the reviewers for their helpful suggestions and constructive reviews. We are encouraged by their positive assessment of our work as novel (zYBJ), creative and original (EvEn), with our core contribution—cross-modal attention instillation—recognized as elegant (zYBJ), effective (zYBJ, wJZj), reasonable (GgBm), and interesting (wJZj). The reviewers acknowledged our strong empirical results (zYBJ, EvEn, wJZj) in both in-domain and out-of-domain scenarios (GgMm) and our comprehensive ablative evaluation (zYBJ, EvEn, wJZj). We are particularly encouraged by Reviewer EvEn's assessment of our work as being "significant for pose-free extrapolative NVS and image–geometry alignment".

We have prepared a **supplementary webpage** (https://e4rks32.github.io/supplementary-webpage/) with additional experiments addressing the reviewers' concerns. These results are also included in our revised manuscript. Key points are summarized below.

&nbsp;

### Key concerns and our responses:
1. **Robustness to geometry prediction errors (Reviewers zYBJ, EvEn, wJZj)**

    Multiple reviewers questioned whether our model is robust to noise and artifacts from external geometry prediction models used to predict sparse geometry for correspondence conditioning. We provide comprehensive experimental validation:

    - **Robustness to noise**: Qualitative and quantitative results under varying Gaussian noise (perturbation) levels (Supplementary webpage **Fig. 4, 6** / Revised paper Table 5, Fig. 10)
    - **Robustness to sparsity**: Qualitative and quantitative results under increased correspondence sparsity (Supplementary webpage **Fig. 5, 6** / revised paper Table 6, Fig. 11)
    - **External model substitution**: Quantitative evaluation when switching external models at inference time, showing minimal performance degradation (Table 3 in response to Reviewer zYBJ)

    These results demonstrate that our model, due to its generative capabilities, possess strong robustness against various error types and levels, effectively preventing error propagation to final outputs.

&nbsp;

2. **Lack of comparison to recent large-scale NVS baselines (Reviewers zYBJ, wJZj, GgMm)**

      Multiple reviewers requested comparisons against recent large-scale NVS models (ZeroNVS, ViewCrafter). We provide additional qualitative comparisons:
    - **In-domain**: Qualitative comparison against VGGT, LVSM, ZeroNVS, and ViewCrafter on RealEstate10k dataset under extrapolative camera poses (Supplementary webpage **Fig. 1** / revised paper Fig. 7)
    - **Out-of-domain**: Qualitative comparison against VGGT, LVSM, and ViewCrafter on unseen, in-the-wild Navi dataset under extrapolative camera poses (Supplementary webpage **Fig. 2** / revised paper Fig. 9)
   - **Quantitative evaluation**: Comparison against LVSM and ViewCrafter on unseen DTU dataset under extrapolative camera poses (Revised paper Table 7)

    These results show that our model achieves superior extrapolative novel view generation with significantly shorter inference time (9s) compared to ZeroNVS (1.5h+) and ViewCrafter (120s+), showing competitive state-of-the-art performance.

&nbsp;

3. **Novelty and contribution (Reviewer wJZj)**

    Reviewer wJZj stated that "integration of diffusion priors to help NVS has been extensively explored in recent work", questioning the novelty of our method. We clarified that our core contributions, especially cross-modal attention instillation (MoAI), are novel and distinct:

    - Unlike previous diffusion-based NVS methods that require expensive per-scene optimization (ZeroNVS, Reconfusion) for robust geometric consistency, we achieve geometric consistency through architectural design (correspondence conditioning)
    - Cross-modal attention instillation enables **geometry supervision to propagate to image** generation (Fig. 3 of our paper), producing geometrically consistent images and aligned geometry in a single feedforward pass
    - Achieves competitive performance with drastically faster inference (9.67s vs. 120s+ for ViewCrafter) while jointly generating scene geometry

&nbsp;

**Additional key updates include:**

- Generalization results on unseen in-the-wild / urban datasets in Fig. 12 of Appendix / **Fig. 3** of our webpage (Reviewer zYBJ)
- Qualitative evaluation of our model’s occlusion/shadow handling at 90°+ extrapolative views in Fig. 13 of Appendix / **Fig. 8** of our webpage (Reviewer zYBJ)
- Quantitative evaluation of replacing our external prediction with a lightweight depth prediction model (Table 4 in response to Reviewer zYBJ)
- Alignment evaluation of predicted novel view geometry in Fig. 14 of Appendix / **Fig. 7** of our webpage (Reviewer EvEn)
- Quantitative ablation of geometry network (Table 1, Reviewer GgBm)
- Quantitative ablation of instilled attention layers (Table 4, Reviewer EvEn)
- Additional elaboration on computation cost and inference runtime in Appendix Sec A.2

---

### Meta-Review · Area_Chair_Xbzi · 2026-01-07

**Summary:**

This paper proposes a diffusion-based warping-and-inpainting framework for pose-free extrapolative novel view synthesis that jointly generates a novel-view image and a corresponding geometry pointmap. The main technical contribution is cross-modal attention instillation (MoAI), which injects the image diffusion U-Net’s attention maps into a parallel geometry diffusion U-Net to improve image–geometry alignment.

Reviewers found the core idea interesting/creative/technically elegant and the empirical results strong in extrapolative settings (notably on DTU and RealEstate10K), with ablations supporting gains from pointmap/mesh conditioning and MoAI. The main concerns before rebuttal were reliance on off-the-shelf geometry/pose predictors and robustness to their errors, missing comparisons to recent large-scale diffusion/view-synthesis baselines, missing runtime/memory reporting, and requests for clearer geometry-branch losses/geometry-side ablations and more detailed MoAI ablations.

Two reviews are already marginally above the acceptance threshold, and the rebuttal directly addresses the common cross-review gaps that drove the two marginally-below ratings: it adds robustness studies for noisy/sparse geometry priors, provides comparisons to recent large-scale NVS baselines (including quantitative DTU extrapolative results for ViewCrafter and LVSM), and reports runtime/memory.

**Reviewer Concerns:**

Addressed by rebuttal:

1. Robustness to geometry prediction errors and correspondence sparsity: additional experiments perturbing the predicted geometry with Gaussian noise and random masking were provided (tables and referenced qualitative figures).

2. External predictor substitution: an additional evaluation swapping the external geometry backbone to DepthAnything v3 was reported (table provided).

3. Comparisons to recent large-scale baselines: added qualitative comparisons (referenced figures) and quantitative DTU extrapolative comparisons against ViewCrafter (1-view) and LVSM (2-view) (tables provided).

4. MoAI ablation depth: a layer-group ablation for which attention layers to instill was added (table provided, with the note that it used abbreviated training during the rebuttal).

Still outstanding:

1. Additional baselines requested in review (e.g., AnySplat; DUSt3R+PnP+splatting): authors did not implement these within rebuttal and commit to include them in the camera-ready.

2. Failure-case taxonomy and deeper diagnostic analysis: partially improved via added qualitative examples and discussion.

**Reviewer Scores:**

zYBJ: 4

EvEn: 6

wJZj: 4

GgBm: 6

---

### Decision · Program_Chairs · 2026-01-26

Accept (Poster)